# Neuro-Symbolic AI for Analytical Solutions of Differential Equations

**Orestis Oikonomou** [1 2 3]  **Levi Lingsch** [1 2 3]  **Dana Grund** [4]  **Siddhartha Mishra** [1 2]  **Georgios Kissas** [5]

## Abstract

Analytical solutions to differential equations offer exact, interpretable insight but are rarely available because discovering them requires expert intuition or exhaustive search of combinatorial spaces. We introduce SIGS, a neuro-symbolic framework for equation-driven closed-form solution discovery. SIGS uses a context-free grammar to generate mathematically valid and physically meaningful building blocks, with a user-specified Ansatz prescribing how these blocks combine, embeds them into a topology-regularised continuous latent manifold, and searches this manifold in two stages: structure selection followed by coefficient refinement using gradient descent, scoring candidates only against the PDE residual and prescribed boundary and initial conditions. This design unifies symbolic reasoning with numerical optimization; the grammar constrains candidate solution blocks to be proper by construction, while the latent search makes exploration tractable and data-free. SIGS is the first neuro-symbolic method to (i) recover analytical solutions for coupled nonlinear PDE systems, (ii) discover equivalent symbolic forms when the grammar lacks the natural primitives, and (iii) produce accurate symbolic approximations for PDEs lacking known closed-form solutions. Overall, SIGS improves over existing symbolic methods by orders of magnitude in both accuracy and runtime across standard PDE benchmarks.

---

[1]Seminar for Applied Mathematics, ETH Zurich, Switzerland [2]ETH AI Center, Zurich, Switzerland [3]IBM Research Europe, Zurich, Switzerland [4]Institute for Atmospheric and Climate Science, ETH Zurich, Switzerland [5]Swiss Data Science Center, ETH Zurich, Switzerland. Correspondence to: Orestis Oikonomou <orestis.oikonomou@ai.ethz.ch>.

*Proceedings of the 43rd International Conference on Machine Learning*, Seoul, South Korea. PMLR 306, 2026. Copyright 2026 by the author(s).

## 1. Introduction

Partial Differential Equations (PDEs) describe the evolution of physical quantities in spacetime. Analytical solutions, closed-form expressions that satisfy the governing equations and conditions, provide far more information than numerical approximations. They reveal intrinsic properties of a system, such as stability, symmetry, and explicit parameter dependencies that explain why systems behave as they do. For this reason, they provide the "gold standard" for benchmarking and physical insight (Ames, 1965; Debnath, 2012). For example, the analytical inversion of the Radon transform remains foundational for CT image reconstruction (Natterer, 2001), soliton solutions of the nonlinear Schrödinger equation specify the shapes that carry intercontinental internet traffic without dispersive degradation (Hasegawa & Tappert, 1973), and analytical approximations form the basis for large-scale engineering and scientific endeavours (Park et al., 1986). In each case, the analytical solution provides instant evaluation and explicit insight into how parameters affect system output. This interpretability is why closed-form solutions remain central to the design of safety-critical applications, despite the availability of computational techniques.

Despite their importance, discovering analytical solutions remains largely manual and thus largely affected by the ingenuity and mathematical insight available to human experts. The core difficulty is combinatorial in essence: a solution is constructed using the correct combination of elementary functions, operators, and structural Ansätze. This design space explodes as the problem complexity increases. The same combinatorial obstacle is central in symbolic regression, which searches symbolic expression spaces to recover formulas or governing laws from data (Schmidt & Lipson, 2009; Udrescu & Tegmark, 2020; Petersen et al., 2019; Biggio et al., 2021; Kamienny et al., 2022; Yu et al., 2025; 2026). Here the task is different: the governing differential equation and prescribed conditions are given, and the unknown is an analytical function that satisfies them. Recent work on solution discovery tries to overcome this complexity bottleneck by augmenting symbolic methods with artificial intelligence. Lample & Charton (2019) trained neural networks for the exact inversion of simple explicit ODEs. Wei et al. (2024) propose SSDE, which uses a recurrent network to generate symbolic candidates under a reinforcement-learning policy

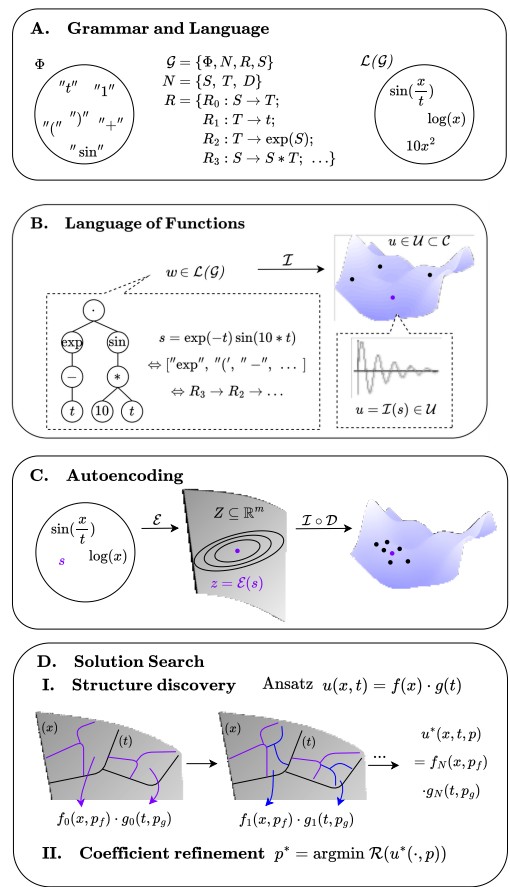

**Figure 1.** Overview over the proposed Symbolic Iterative Grammar Solver (SIGS). **A.** Terminal symbols $\Phi$ and rules $R$, together with non-terminals $N$ and starting symbol $S$, form the grammar $\mathcal{G}$ which generates the mathematical expressions in the library $\mathcal{L}(\mathcal{G})$. **B.** Each expression $w \in \mathcal{L}(\mathcal{G})$ is identified with a function $u$ in the finite set of candidate functions $\mathcal{U}$. **C.** The encoder $\mathcal{E}$ and decoder $\mathcal{D}$ of the Grammar Variational Autoencoder (GVAE, (Kusner et al., 2017)) embed the finite $\mathcal{L}(\mathcal{G})$ into the continuous latent space $Z$. **D.** Given a differential equation and system conditions, a structure search is performed over $z \in Z$ using iterative clustering, followed by a separate optimization of the constants in the final structure, optimizing for lowest residual $\mathcal{R}$ of the corresponding candidate function $u = (\mathcal{I} \circ \mathcal{D})(z) \in \mathcal{U}$.

constrained by governing equations. Cao et al. (2024) employ transfer learning to lift genetic-programming results from one-dimensional problems to higher dimensions (HD-TLGP), addressing difficulties found in initial efforts at solution discovery (Tsoulos & Lagaris, 2006; Seaton et al., 2010; Kamali et al., 2015; Boudouaoui et al., 2020). Consequently, existing methods cluster at two extremes, namely unconstrained search, which suffers from combinatorial explosion and sensitivity to the initial guess, or narrow pretraining, which biases discovery to a limited class of equations. A principled middle ground, aiming to constrain the search to mathematically and physically admissible solutions, while also generalizing to different PDE classes and conditions, is still not available.

We address this gap by proposing the **Symbolic Iterative Grammar Solver (SIGS)**, a neuro-symbolic framework that casts analytical PDE solution discovery as a *hierarchical, grammar-guided composition* of analytic atoms. At the top level, an Ansatz specifies the structural form of candidate solutions by a combination of atoms drawn from a formal grammar (Hopcroft & Ullman, 1979), where elementary functions serve as terminals and operations such as addition and exponentiation act as production rules. This formalism generalizes classical construction techniques: traditionally, a practitioner selects an Ansatz and manually searches for atoms that satisfy the PDE and boundary/initial conditions; SIGS automates this by treating grammatically admissible expressions as candidates and evaluating their fit against the governing equations and prescribed conditions. To navigate the space of candidates efficiently, SIGS embeds grammar-generated atoms into a continuous latent manifold using a Grammar Variational Autoencoder (Kusner et al., 2017), augmented with a topological regularisation that makes latent neighborhoods decode smoothly to valid expressions. This transforms combinatorial tree search into geometric exploration of promising regions rather than enumeration of symbolic trees. The same grammar-generated corpus and trained manifold are reused across all benchmarks, with only the high-level Ansatz adapting per problem. For each new PDE, SIGS searches this fixed manifold in two stages: Stage I identifies symbolic structures with low residual, and Stage II refines the remaining numerical constants by gradient descent to yield exact or high-precision analytical solutions.

Our contributions are summarized as follows:

- We propose SIGS, a grammar-based framework that casts analytical PDE solution discovery as a hierarchical composition of atoms through an Ansatz. A fixed grammar and latent manifold are reused across the tested PDE classes, while the Ansatz defines the admissible assembly family for each problem.

- SIGS is equation-driven: it *requires no simulation data and no per-problem retraining*. Candidates are scored directly by the PDE residual and boundary and initial conditions.

- SIGS improves over recent symbolic PDE baselines by *orders of magnitude in accuracy and runtime on benchmarks*, recovering known analytical solutions up to symbolic equivalence and numerical tolerance.

- SIGS is the first method to *handle coupled nonlinear PDE systems, PDEs without known closed-forms, controlled primitive removal, and over-specified Ansätze.*

- We introduce a topology-aware latent regulariser that *improves off-training decode efficiency* relative to a vanilla GVAE (Table 7).

## 2. Method

**Problem setup**    We consider the generic form of a time-dependent partial differential equation (PDE) as (Molinaro et al., 2024),

$$
\begin{aligned}
\mathbb{D}(u) &= \mathbf{f}, & \forall(\mathbf{x}, t) \in \Omega \times [0, T], \\
u(\mathbf{x}, 0) &= u_0(\mathbf{x}), & \forall \mathbf{x} \in \Omega, \\
\mathbb{B}[u](\mathbf{x}, t) &= g, & \forall(\mathbf{x}, t) \in \partial\Omega \times [0, T],
\end{aligned}
\tag{1}
$$

where $\Omega \subset \mathbb{R}^d$ is the spatial domain, $u \in \mathcal{U} \subseteq \mathcal{C}(\Omega \times (0, T))$ is the space-time continuous solution, $\mathbf{f} \in \overline{\mathcal{U}}$ is a forcing term, $u_0 \in H^s(\Omega)$ is an initial condition, $\mathbb{B}[u](\mathbf{x}, t)$ denotes the boundary conditions, and $\partial\Omega$ is the boundary of the domain. The differential operator includes PDE parameters $\xi \in \mathbb{R}^{d_\xi}$, time and higher order derivatives, $\mathbb{D}(u) = \mathbb{D}(\xi, u, \partial_t u, \partial_{tt} u, \nabla_{\mathbf{x}}, \nabla_x^2, \cdots)$. Equation 1 represents a very general form of differential equations by setting $u = u(t)$, we recover general ODEs. Setting $u = u(x)$ enables us to recover time-independent PDEs from the same formulation. We call the collection of $\mathbf{f}, \mathbb{B}[u]$, and $u_0$ the *system conditions* that need to be specified to solve a given PDE. We define the *symbolic form* of a PDE as:

$$
\mathcal{S}(u) = \mathbb{D}(u) - \mathbf{f}, \quad \forall u \in \mathcal{U}.
$$

We formulate solving PDEs as an iterative computational process where, given a domain discretization, boundary and initial conditions, and the symbolic form of the PDE

$$
(\Omega, \mathbb{B}[u], u_0, S) \xrightarrow{\mathcal{D}(z)} u.
$$

The method searches for a parameterization $z$ of $u_z \in \mathcal{U}$ that minimizes the loss,

$$
\mathcal{R}(u_z) = \|\mathcal{S}(u_z)\|^2 + \|u_z(0, x) - u_0\|^2 + \|\mathbb{B}[u_z] - g\|^2, \tag{2}
$$

where we generally use equal weighting between the residual terms. An analytical *solution* is recovered when $\mathcal{R}(u_z) = 0$ and an *approximate* analytical solution if $0 < \mathcal{R}(u_z) \ll 1$.

**Grammar Construction.**    Analytic expressions are commonly represented as trees, with internal node labels denoting unary or binary expressions (e.g. "sin", "+") and leaves denoting constants or variables. However, generating such trees by randomly sampling can produce many syntactically ill-formed expressions (Virgolin & Pissis, 2022; Kissas et al., 2024). To alleviate this issue, we use a *Context-Free Grammar (CFG)* (Chomsky, 1956; Hopcroft & Ullman, 1979) as a principled way to generate exactly the classes of atoms admitted by an Ansatz. The plain CFG is defined as $\mathcal{G} = \{\Phi, N, R, S\}$, where $\Phi$ is the set of terminal symbols, $N$ is the set of non-terminal symbols and $\Phi \cap N = \emptyset$, $R$ is a finite set of production rules and $S \in N$ is the starting symbol. Each rule $r \in R$ is a map

$\alpha \to \beta$, where $\alpha \in N$, and $\beta \in (\Phi \cup N)^*$ (see Fig. 1A). A language $\mathcal{L}(\mathcal{G})$ is defined as the set of all possible terminal strings that can be derived by applying the production rules of the grammar starting from $S$, or all possible ways that the nodes of a derivation tree can be connected starting from $S$ as $\mathcal{L}(\mathcal{G}) = \{w \in \Phi^* \mid S \to^* w\}$, where $\to^*$ implies $T \geq 0$ applications of rules in $R$. Each expression is equivalently represented by the string $w$ (as a sequence of symbols), by the list of rules applied to generate $w$ from $S$, and by a *derivation tree* that represents the syntactic structure of string $w \in \mathcal{L}(\mathcal{G})$ according to grammar $\mathcal{G}$. We define an interpretation map $\mathcal{I} : \mathcal{L}(\mathcal{G}) \to \mathcal{U}$, which assigns to each syntactic expression $w \in \mathcal{L}(\mathcal{G})$ semantic meaning in terms of a function $u_w : D \to \mathbb{R}$. The set of all functions represented by the grammar is $\mathcal{U}(\mathcal{G}) = \{u_w : D \to \mathbb{R} \mid u_w = \mathcal{I}(w), w \in \mathcal{L}(\mathcal{G})\}$. We refer to $u_w$ as $u$ in the future to simplify the notation.

**Compositional Ansatz and Atoms.**    The computational Ansatz is a critical foundation of numerical and symbolic solutions methods alike. Finite elements, for example, assume the structure of the solution to be $u(x) = \sum_i u_i \phi_i(x)$, where $\phi_i$ is a user-prescribed basis function. Similarly, neural networks assume $u(x) = L_N \circ ... \circ L_1(x)$ where $L(x) = \sigma \circ (Wx + b)$, where $\sigma$ and $N$ are a user-defined activation and number of layers. Symbolic approaches also rely on this paradigm, e.g. separation of variables considers an Ansatz $u(x_1, x_2) = \sum_i X_1(x_1) X_2(x_2)$, and the neuro-symbolic methods SSDE and HD-TLGP define an Ansatz $u(x_1, ..., x_N) = f_N(x_N, f_{N-1}(N_{N-1}, ..., f_1(x_1)))$ and $u(x_1, ..., x_N) = f_1(x_1)...f_N(x_N)$, respectively, where $f_i$ is a function class defined by the dictionary.

In this work, we generalize this concept, equipping SIGS with the flexibility to handle generic forms as well as specific structures which incorporate domain specific knowledge and respect boundary conditions and the operator class. As a result, SIGS can restrict the search over previously intractable spaces to regions with a high probability of containing the true form of the analytical solution.

When solving a PDE using SIGS, the user can specify a structural Ansatz $F$ that guides the composition of the proposed solution. For example, one can specify spatiotemporal separability as $u(x, t) = \sum_{j=1}^{K} a_j T_j(t) \phi_j(x)$, leaving the spatial eigenfunctions $\phi_j$ and temporal factors $T_j$ to be chosen by SIGS. In addition to $\phi_j$ and $T_j$, the user may include atoms that encode physical mechanisms at the expression level, such as transport phases, $kx - \omega t$; viscous shock profiles, $\tanh((x_0 + x - ct)/\nu)$; or other motifs known to describe the dynamics of interest exactly or approximately. Localized atoms such as Gaussians can also be included to capture spatially confined phenomena. The Ansatz may include hybrid factors that mix space and time, allowing $u(x, t) = \sum_{j=1}^{K} a_j T_j(t) \phi_j(x) \psi_j(x, t)$ which re-

laxes separability while retaining a controlled, interpretable composition.

To efficiently search the hypothesis space, we embed atoms (sub-trees) instead of primitives (unary, binary operators, reals, and variables) to decrease the combinatorial complexity of the problem. In the full Ansatz generality, the solution construction could be performed by considering a number of arbitrary combinations between atoms. This approach would result in a combinatorial explosion, partially losing the benefit of considering atoms. For this reason, we assume that the solutions can be described exactly (or sufficiently well) by the chosen Ansatz. To include the Ansatz into the grammar, we denote by $A : \{\mathcal{L}(\mathcal{G})\}^L \to \mathcal{L}(\mathcal{G})$ the assembly map that composes the individual components into the final solution following the Ansatz. This restricted function class is obtained by activating only those nonterminals and production rules that implement the user's Ansatz and its permitted atom categories, and by enforcing the assembly production dictated by $A$. The Ansatz thus induces a restriction on the language $\mathcal{L}_A(\mathcal{G}) = \{A(w^1, ..., w^L) : w^c \in \mathcal{L}_c(\mathcal{G})\}$ for the component classes $c$ required by the Ansatz. In all cases, $A$ realizes the user's choice by assembling requested categories into a single symbolic candidate that is then scored by the PDE residual. In summary, the Ansatz specifies which families of atoms and couplings are admissible, the CFG generates those atoms and couplings, and the interpretation map turns each derivation into a candidate function over which SIGS optimizes the PDE residual.

**Grammar Variational Autoencoders.** To make the search more efficient, we embed $w \in \mathcal{L}_A(\mathcal{G})$ into a low-dimensional continuous manifold using a Grammar Variational Autoencoder (Kusner et al., 2017). The encoder is defined as $q_\phi(z|w)$ and the decoder $p_\theta(w|z)$, for $z \in Z$ and $w \in \mathcal{L}_A(\mathcal{G})$. The GVAE is trained by minimizing the objective:

$$\mathcal{L} = \mathcal{L}_{\text{recon}} + \gamma \, \text{KL}(q(z|w) \| p(z)),$$

where $\mathcal{L}_{\text{recon}}$ is the cross-entropy loss between the predicted and the baseline grammar rules, and $\text{KL}(q(z|w) \| p(z))$ is the KL divergence between the encoder and the prior distributions. Training the GVAE *does not require numerical data, only expressions* $w \in \mathcal{L}$. In practice, the model is trained using grammar masks and one-hot encoding matrices denoting which rules are allowed, which are used, and in what order for generating $w$. Training is one-off across all problems and consists of 23,682 expressions, 51 rules and fixed hyperparameters (for more details see Appendix A and B).

**Geometry regularisation.** When sampling the latent manifold, candidates often land in regions with little or no support from the training distribution, or get trapped in topological artifacts of the latent space, and in both cases the decoder produces degenerate outputs. Therefore, to convert the GVAE latent into a topology-regularised continuous latent manifold, we impose a geometry-aware regulariser that constrains the search inside a data-supported enclosure, removes small topological artifacts at the working resolution, and smooths the decoder so that small latent moves produce predictable changes in the decoded function. We augment the GVAE objective with three regularisers (details in App. B.3). A convex-enclosure loss $\mathcal{L}_{\text{Hull}}$ that discourages latents from leaving the data-supported region estimated from training codes (Gonzalez, 1985; Rockafellar, 2015); a persistent-homology loss $\mathcal{L}_{\text{ph}}$ that suppresses small spurious loops/gaps in the latent cloud at a fixed working scale (Edelsbrunner & Harer, 2010); a decoder-smoothness loss $\mathcal{L}_{\text{smooth}}$ that penalizes large second-order changes in the decoder, so nearby latents decode to predictably similar functions (Hutchinson, 1989). We combine these losses with the reconstruction and the KL loss to define the regularised loss of the TGVAE (Topological Grammar Variational Autoencoder):

$$\mathcal{L} = \mathcal{L}_{\text{recon}} + \gamma \, \text{KL}(q(z \mid w) \| p(z)) + \mathcal{L}_{\text{topo}},$$
$$\mathcal{L}_{\text{topo}} = \mathcal{L}_{\text{Hull}} + \mathcal{L}_{\text{ph}} + \mathcal{L}_{\text{smooth}}$$

The empirical effect on decoder validity is reported in Appendix B.3 and Table 7.

**Solution Discovery.** The solution discovery is split into two stages (see Fig. 1D, and details in App. C): In the structure search, we iteratively explore the latent space for a candidate function included in the structural Ansatz while minimizing the PDE residual, and then optimize its numerical constants in a separate stage. For searching, we consider a deterministic encoding $\mathcal{E}(w) = \mu_\phi(w) \in Z$ and a decoding $\mathcal{D} : Z \to \mathcal{L}_A(\mathcal{G})$ obtained by the argmax decoding under the grammar mask. Composing with $\mathcal{I}$, we have $\mathcal{I} \circ \mathcal{D} : Z \to \mathcal{U}_A(\mathcal{G})$, so each $z \in Z$ corresponds to a function $u = \mathcal{I}(\mathcal{D}(z)) \in \mathcal{U}_A(\mathcal{G})$. Let $\tau : \mathcal{L}(\mathcal{G}) \to \mathcal{T}$ be a semantic map that assigns tags, e.g. variables, deterministically computed from the parse tree $w$. The map is computed once after training, and used for any downstream solution problem. For a given differential equation, we choose the admissible tag set, e.g. any function with $x, y$ arguments, and restrict the search to the type-constrained latent subspace $Z' = \{z \in Z : \tau(\mathcal{D}(z)) \in \mathcal{T}' \subseteq \mathcal{T}\}$. Let $\kappa : Z' \to \{1, ..., m\}$ be a clustering map in the latent space and denote the clusters $C_j = \kappa^{-1}(j)$. We cluster a given subspace based on $z \in Z'$, and then solve a discrete selection problem to choose the cluster that contains the most promising solution forms for each $T$ and $\phi$, $j^* = \arg \min_{1 \leq j \leq m} [\inf_{z \in Z_j \subset C_j} \mathcal{R}(\mathcal{D}(z))]$, where $Z_j$ can be constructed by either only the expressions from the training set that fall in $Z'$ or the expressions together with samples from the generative model. Within the best cluster $C_{j^*}$, a global

latent search is performed:

$$z^* = \arg \min_{z \in C_{j^*}} \mathcal{R}(\mathcal{D}(z)),$$

either by a global optimizer or by iterative clustering, performing discrete selection, and sampling from the most promising cluster until $\mathcal{R}(\mathcal{D}(z))$ drops below a threshold. The solution takes a parametric form $u_z(\cdot, p)$, including constants $p$ that are only represented in the grammar with limited precision. Thus, we perform a parameter refining step. We consider a gradient based method (Adam, Kingma & Ba, 2014), and minimize the loss until $\mathcal{R}(u) \leq \varepsilon_{\mathrm{tol}} = 10^{-8}$ is reached (this tolerance is used for all Stage II runs unless stated otherwise). The loss $\mathcal{R}(u)$ is augmented here by the hull loss $\mathcal{R}'(u) = \mathcal{R}(u) + \mathcal{L}_{\mathrm{hull}}$ to penalize whichever latent falls out of the hull defined during training. Algorithm 1 summarizes the pipeline; full sub-algorithms and search settings are in Appendix C.

---

**Algorithm 1** SIGS: high-level structure

---

1: **Input:** PDE residual $\mathcal{R}$, Ansatz $A$, encoder/decoder $(\mathcal{E}, \mathcal{D})$, atom corpus $\mathcal{L}$
2: **Output:** symbolic candidate $u^\star$ with $\mathcal{R}(u^\star) \leq \varepsilon$
3: After GVAE training encode-decode the corpus and compute tags for every atom
4: For each Ansatz slot, keep tag-admissible atoms and cluster their encodings
5: *// Stage I: structure search*
6: **repeat**
7:     Sample one atom per cluster per slot; assemble cross-slot combinations via $A$ and score each assembled candidate by $\mathcal{R}$
8:     Identify the winning cluster per slot from the lowest-residual combination; repartition each into sub-clusters
9: **until** structural residual below threshold or budget exhausted
10: *// Stage II: numerical refinement*
11: Refine constants in selected $w^\star$ via Adam multi-start
12: **return** $u^\star$

---

## 3. Experiments and Results

We conduct comprehensive experiments to evaluate SIGS' performance against state-of-the-art symbolic, neural, and numerical solvers. Our evaluation comprises: (i) cross-validation on benchmarks sourced from the literature, (ii) comparison on benchmarks with known analytical solutions, (iii) PDEs without known analytical solutions, (iv) systems of PDEs, and (v) ablations of primitives, atoms, latent search, topology-aware regularisation, and grammar completeness. Across all experiments, SIGS uses a single fixed grammar, a 23,682-expression atom corpus, and one trained GVAE/TGVAE manifold; only the PDE residual and high-level Ansatz are specified per problem. Details on the grammar, corpus, one-time TGVAE training, full

Ansätze, and search-refinement algorithm can be found in Appendix A–C. In contrast, the symbolic baselines require problem-specific primitive dictionaries or knowledge bases as detailed in Appendix E.

**Experimental Setup.** Our benchmark suite comprises ten PDEs of hyperbolic, parabolic, and elliptic families, including both scalar equations and coupled systems. Three scalar problems admit known analytical solutions: viscous Burgers, 1D Diffusion and 2D Damped Wave equations. For the case with no known analytic solution, we consider three Poisson problems with superposition of different numbers of Gaussian source terms to test the approximation capabilities of the method. To evaluate SIGS on more challenging scenarios, we consider three problems: the Korteweg-de Vries (KdV) equation to test robustness under grammar misspecification (missing primitive functions), and two coupled nonlinear systems, the 2D Shallow Water Equations (SWE) and 2D Compressible Euler (CE) equations, to demonstrate SIGS's capability to discover approximate analytical solutions for systems of PDEs.

We compare against two recent symbolic discovery methods: HD-TLGP (Cao et al., 2024), and SSDE (Wei et al., 2024). Both methods sample discrete trees by combinatorially combining elements of a user-defined dictionary. Moreover, HD-TLGP considers an Ansatz where the solution is separable in dimension, e.g. $u(x, y) = f(x)g(y)$, as well as solution structure in one dimension as prior knowledge. SSDE considers a recursive single-variable decomposition Ansatz, e.g. $u(x, y) = g(x, f(y, c))$ and couples reinforcement-learning with a hierarchical approach that resolves each recursion depth sequentially. Both methods search for expressions that minimize physics-aware losses similar to $\mathcal{R}(u)$. For HD-TLGP we use its vanilla primitive-level dictionary setup, which is the comparison HD-TLGP was published with. To separately test whether SIGS's gain comes from its atom-level vocabulary or its latent-manifold search, we additionally run HD-TLGP's tree search over the same atom-level components SIGS uses, and report that result as a controlled ablation in Section (v). For SSDE, we tailor the dictionary of terms for each problem to contain only the primitives and variables contained in the solution. For example, if $u(x) = \sin(\pi x) + \cos(\pi y)$, the dictionary contains only $\sin, x, y, +, \pi, \cos$ and integers. In this way, we show that for sophisticated search methods, if the dictionary considers primitives instead of atoms, the method cannot find an admissible solution when we consider complex problems. The complete primitive specifications appear in Appendix E.2.

Neural baselines (PINNs (Raissi et al., 2019), FBPINNs (Moseley et al., 2023)) and numerical solvers (FEniCS; Alnæs et al., 2015, ; see details in Appendix F.4) are included for reference. For the case of a PDE without a known solution, we compare also against the Computer

*Table 1.* We compare the accuracy, in terms of relative $L_2$ error against the exact solution, of SIGS and baselines on a collection of PDEs presented in the HD-TLGP and SSDE papers.

| Problem (method) | Original Method | SIGS (ours) |
|---|---|---|
| Poisson (HD-TLGP) | $4.36 \times 10^{-4}$ | **exact solution** |
| Advection (HD-TLGP) | $1.01 \times 10^{-2}$ | **exact solution** |
| Wave (SSDE) | $1.04 \times 10^{-16}$ | **exact solution** |

Algebra System (CAS) Mathematica's `DSolveValue` and an extended-reasoning LLM check; prompt templates and evaluation details are provided in Appendix I. For the Poisson-Gauss problems, no analytical solution is available. Therefore, we assume FEniCS with P4 elements on a $128 \times 128$ mesh as the ground truth. We perform a mesh convergence study to confirm convergence of the solution at the chosen resolution. Complete problem specifications, analytical solutions, and discovered symbolic forms relevant to all problems in the suite appear in Appendix D, accompanied by additional figures in Appendix F.3.

**(i) Cross-validation on literature's benchmarks.** First, we test SIGS on a subset of problems considered by Cao et al. (2024) and Wei et al. (2024) to show how combining grammar-generated atoms with adaptive search is more accurate than alternative approaches. For this purpose, we chose one-dimensional Poisson and Advection PDEs (HD-TLGP), and a two-dimensional Wave PDE (SSDE), as summarized in Appendix D.1. We consider the same problem specification, that is, the domain, boundary, and initial conditions, for the comparison. We impose the same high-level Ansatz to SIGS as HD-TLGP, e.g. $u(x,t) = g(x)f(t)$. The results are presented in Table 1. While the baseline approaches achieve high accuracy (HD-TLGP: $4.36 \times 10^{-4}$ for Poisson, and $1.01 \times 10^{-2}$ for Advection; SSDE: $1.04 \times 10^{-16}$ for Wave), SIGS recovers equivalent symbolic forms on all problems with machine precision accuracy, as constants such as $\pi$ are grammar terminals and are therefore represented symbolically and verified to float64 tolerance.

**(ii) Complex PDEs with known solutions.** What makes the following collection of experiments complicated is not only that the solution contains many terms, but also that the method needs to find solutions that are very precise. For example, even if an algorithm discovers a solution that describes a viscous shock for the Burgers equation, slight imprecision in the location of the shock will result in a very large relative $L_2$ error against the exact solution. This is also true for the damped wave, as the problem is sensitive to the coefficients governing the diffusion time. For SIGS, we consider the solution Ansätze in Table 2, with full problem specifications in Appendix D.1. Table 2 fixes the high-level assembly family; the grammar-generated atoms filling each slot are selected by latent search. We present the results in Table 3. We observe that SIGS recovers expressions equivalent to the analytical solutions, achieving machine

*Table 2.* Ansatze used for each benchmark family (with learned coefficients $a$ or $a_i$ where applicable).

| Problem | Ansatze |
|---|---|
| Burgers | $u(x,t) = a_0 + a_1\,\psi(x,t)$ |
| Wave | $u(x,y,t) = a\,\phi^1(x)\,\phi^2(y)\,T(t)$ |
| PG-2/3/4 | $u(x,y) = \sin(\pi x)\sin(\pi y)\sum_{i=1}^{K} a_i\,\psi_i(x,y)$ |
| Advection | $u(x,t) = a\,\psi(x,t)$ |
| Diffusion | $u(x,t) = \sum_{i=1}^{3} a_i\,\phi_i(x)\,T_i(t)$ |
| Damped wave | $u(x,y,t) = a\,\psi(x,y,t)\,T(t)$ |
| KdV | $u(x,t) = \sum_{k=0}^{2} a_k\,\psi(x,t)^k$ |
| Poisson | $u(x,y) = a^1\phi^1(x) + a^2\phi^2(y)$ |
| | $\rho(x,y,t) = \psi^1(x,y,t)\,\psi^2(x,y,t)\,T(t),$ |
| Shallow Water | $u(x,y,t) = \psi^3(x,y,t)\,\rho(x,y,t),$ |
| | $v(x,y,t) = \psi^4(x,y,t)\,\rho(x,y,t).$ |
| | $\rho(x,y) = \exp\!\Big(\sum_{i=1}^{6} f_i(x,y)\Big),$ |
| | $u(x,y) = \sum_{i=1}^{6} g_i(x,y),$ |
| Compressible Euler | $v(x,y) = \sum_{i=1}^{6} h_i(x,y),$ |
| | $p(x,y) = \exp\!\Big(\sum_{i=1}^{6} k_i(x,y)\Big).$ |

*Table 3.* Comparison of methods on PDEs with known analytical solutions. Reported are relative $L^2$ errors in percent. SIGS errors correspond to functional recovery up to symbolic equivalence and float64 verification tolerance.

| PDE Problem | SIGS | HD-TLGP(atoms) | HD-TLGP(vanilla) | SSDE | PINNs | FBPINNs | FEniCS |
|---|---|---|---|---|---|---|---|
| Burgers | $6.64 \times 10^{-14}$ | 2.04 | 35.68 | 45.62 | 6.09 | 28.26 | $8.69 \times 10^{-3}$ |
| Diffusion | $7.16 \times 10^{-13}$ | 33.34 | 79.73 | $5.87 \times 10^3$ | 2.56 | 55.54 | $2.26 \times 10^{-3}$ |
| Damped wave | $1.22 \times 10^{-13}$ | 423.30 | 178.77 | $1.19 \times 10^3$ | 5.56 | 71.36 | $2.28 \times 10^{-2}$ |

precision on all problems with relative errors ranging from $6.64 \times 10^{-14}$ to $1.22 \times 10^{-13}$. The discovered expressions match analytical forms up to symbolic equivalence and numerical precision, see Appendix F.2.

Both HD-TLGP and SSDE fail to find an accurate solution within the time budget, see Appendix F.2. HD-TLGP, searching from its vanilla primitive-level dictionary, produces relative $L_2$ errors in the range of 35.68–178.77%. SSDE produces errors in the range of 45.62–5870% even though the primitives are tailored for each problem. We hypothesize that SSDE fails because the reinforcement-learning algorithm must find a small subset of candidate expressions with low residual while avoiding high-residual and invalid regions, a sparse-reward search problem. These results indicate that sophisticated optimizers fail when the dictionary lacks elements that support aggressive and adaptive exploration of the candidate-solution space. The ablations below separate this point further: atoms are necessary for admissible search, while SIGS's latent-manifold search is what makes structure selection tractable. Neural methods achieve moderate accuracy (2.56–6.09%), while numerical solvers (FEniCS) present very accurate field approximations. A visual comparison of the predictions of different methods is provided in Figure 2.

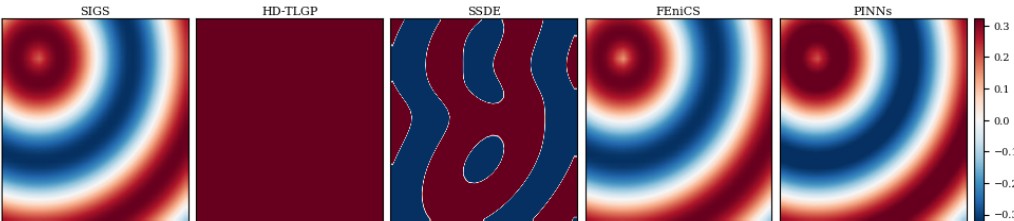

*Figure 2.* Comparison of different methods for solving the damped wave equation at $t = 2.5$. All methods show the same physical domain $x, y \in [-8, 8]$ with wave center at $(-5, 5)$. Parameters: $k = 0.5$, $\omega = 0.4$, $\alpha = 0.45$.

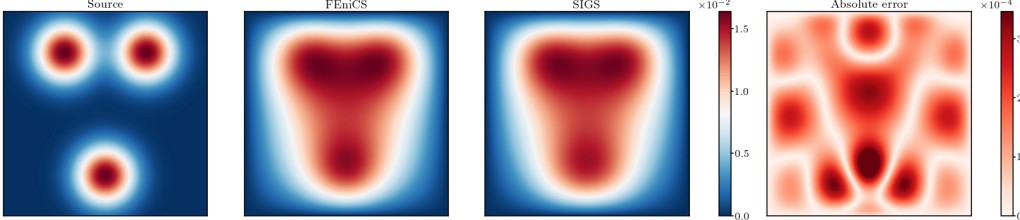

*Figure 3.* (a) source term $F(x, y)$ for the Poisson–Gauss problem; (b) finite-element solution $u_h$ (FEniCS); (c) symbolic approximation $u_{\text{sigs}}$ (SIGS); (d) absolute error $|u_h - u_{\text{sigs}}|$

**(iii) Approximation for unknown solutions.** For the PDEs considered so far, we manufactured, and therefore had access to, the exact solution. This allowed us to make informed decisions about the form of the Ansatz and verify that SIGS identifies compact atoms consistent with the analytical structure. In this example, we test how well SIGS and the baselines approximate the solution of a PDE where no closed-form solution is known. For this case, we consider the Poisson equation with Gaussian forcing terms (Appendix D.1). The forcing is localized, and the Dirichlet Poisson solution admits the Green representation $u(x) = \int_\Omega G(x, \xi) f(\xi)\, d\xi$; localized radial-basis atoms therefore provide a generic approximation mechanism for the Green-smoothed response. For SIGS we choose the generic superposition Ansatz $u(x, y) = \sum_{j=1}^{N} \phi_j(x, y)\psi_j(x, y)$, implemented with a standard homogeneous-Dirichlet mask as $u(x, y) = \sin(\pi x)\sin(\pi y)\sum_{j=1}^{N} a_j \psi_j(x, y)$, with $N = 3, 4, 8$ for PG-2, PG-3, and PG-4, respectively. We present the results for all methods in Table 4, while in Figure 3, we provide a visual comparison of the solutions from FEniCS and SIGS for PG-3. SIGS achieves 1–3% relative $L_2$ errors and returns readable symbolic expressions exposing localized centers, widths, amplitudes, and boundary-compatible structure. HD-TLGP fails to find a solution within the time budget, generating errors of $98.94\%$ on PG-2 and exceeding $10^7$ on PG-3 and PG-4. SSDE achieves errors in the range 58–70%. The results support that successfully discovering useful symbolic approximations requires a combination of structured atoms and a global-local optimization algorithm which simultaneously explores the search space and discovers precise arguments. Moreover, Table 5 shows how the SIGS is practically viable as the solutions are found in seconds to minutes. To contextualize the difficulty of this benchmark suite, we additionally evaluated

*Table 4.* Approximation on Poisson-Gauss problems without analytical solutions. Reported are relative $L^2$ errors in percent against FEniCS references.

| Problem | SIGS | HD-TLGP(atoms) | HD-TLGP(vanilla) | SSDE |
|---------|------|----------------|------------------|------|
| PG-2 | **2.66** | 200.9 | 98.94 | 69.29 |
| PG-3 | **1.54** | NaN | $5.61 \times 10^7$ | 69.64 |
| PG-4 | **1.05** | NaN | $5.45 \times 10^7$ | 58.70 |

*Table 5.* Wall-clock time (CPU) to the same residual threshold $\varepsilon$ used by the baselines, so all methods are compared on equal terms. SIGS reports *time-to-$\varepsilon$*; others report *time-at-termination*. Notation: $\checkmark$ reached $\varepsilon$; $\dagger$ hit budget / failed to reach $\varepsilon$. HD-TLGP budget: 20 generations, SSDE budget: 25 generations.

| Problem | SIGS$\checkmark$ | HD-TLGP(atoms)$\dagger$ | HD-TLGP(vanilla)$\dagger$ | SSDE$\dagger$ | PINNs$\dagger$ | FEniCS$\checkmark$ |
|---------|------|------|------|------|------|------|
| Burgers | 11.62 | **14375.5** | **12057.2** | 393.55 | 8.82 | 2.18 |
| Diffusion | 14.67 | **11560.6** | **10909.3** | 485.73 | 121.69 | 1.38 |
| Damped wave | 8.95 | **5319.5** | **2227.7** | 379.17 | 29.5 | 3.35 |
| PG-2 | 90.4 | **10944.9** | **5442.7** | 704.53 | n/a | 19.32 |
| PG-3 | 111.0 | **7207.0** | **5836.3** | 664.28 | n/a | 6.91 |
| PG-4 | 83.4 | **8731.6** | **4850.4** | 751.56 | n/a | 3.35 |

Mathematica's `DSolveValue` and an extended-reasoning LLM check. `DSolveValue` frequently fails or returns non compact infinite-series/Green representations, while the LLM recovers the manufactured Burgers profile but violates the homogeneous Dirichlet boundary conditions on the Poisson-Gauss equation. Full prompts, outputs, and metrics are provided in Appendix I.

**(iv) Coupled nonlinear PDE systems.** We extend SIGS to coupled nonlinear systems, a regime that lies outside the published scope of the symbolic baselines (HD-TLGP, SSDE), which fail due to combinatorial explosion. Reformulating these methods to handle coupled $(\rho, u, v)$ or $(\rho, u, v, p)$ systems would require modifying their core search and assembly logic, which is beyond a fair head-to-head comparison. We test on the 2D Shallow Water Equa-

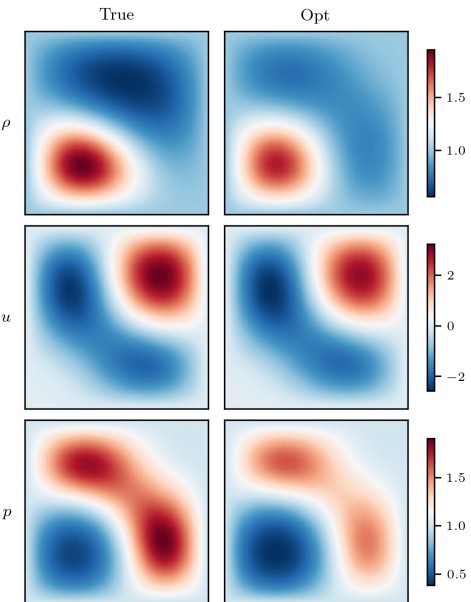

*Figure 4.* Compressible Euler manufactured fields and SIGS-refined fields. Each row corresponds to one state variable $(\rho, u, p)$ ($v$ omitted for space), and columns show the manufactured reference and the SIGS-refined expression. The experiment evaluates coupled manufactured-field structural recovery: SIGS captures the dominant spatial structure and magnitude across the four-field nonlinear system, although coefficients are not recovered exactly.

tions (SWE), with Dirichlet boundary conditions, and steady Compressible Euler (CE) equations with periodic boundary conditions, constructing Ansätze that capture inter-field dependencies (e.g., setting $u, v \propto \rho$ for SWE, and using Fourier-exponential forms for $\rho, p$ in CE; results are summarized in Table 37 and for details see Appendix H). SIGS solves SWE with relative errors $< 0.05\%$ in roughly 3 minutes. CE is a manufactured-reference structural recovery problem, where the analytical reference fields are known, and SIGS recovers the dominant Fourier/exponential field structure with $\approx 10\%$ relative error. The recovered solutions shown in Figure 4 closely match qualitative properties of the manufactured fields, demonstrating that grammar-guided latent search extends to multi-physics couplings.

**(v) Ablation: search space, topology, priors, and grammar completeness.** To isolate the contribution of each SIGS component, we ablate the pipeline one design choice at a time. The ablations follow the order in which candidates are constructed and searched: the granularity of the symbolic vocabulary, the search geometry over that vocabulary, the topology-aware regularisation of the latent manifold, the robustness of the high-level admissible priors, and the completeness of the grammar's primitive set.

**Primitives vs. Atoms.** We first remove atoms and attempt to start from primitive-level CFG proposals. For the damped wave PDE with Ansatz $u(x, y, t) = \sum_{j=1}^{N} \psi_j(x, y, t)\phi_j(x)T_j(t)$, we sample $\phi, T, \psi$ uniformly over grammar rules. If a sampled function has the correct arity, e.g., $\psi$ contains $x, y, t$, we count it as admissible. Out of 50,000 primitive-level proposals, only 133 are admissible, and the best admissible expression has $\approx 366\%$ relative $L^2$ error. Thus the atom vocabulary is not just a convenience: without component-level atoms, the search rarely reaches candidates on which adaptive optimization can start.

**Search geometry.** To isolate the role of the latent manifold from the role of the atom vocabulary, we run HD-TLGP's tree search over the same atom-level components SIGS uses. Even with identical symbolic ingredients, the tree search remains at $2.04\%$, $33.34\%$, and $423.30\%$ error on Burgers, Diffusion, and Damped Wave, while full SIGS reaches $10^{-13}$-scale errors. Atoms are necessary but not sufficient, the latent manifold is what makes the search tractable. Table 6 summarizes these two axes together: primitives versus atoms, and atoms with discrete tree search versus atoms with latent search.

*Table 6.* Ablation axes for vocabulary granularity and search geometry. Task columns report relative $L^2$ errors in percent, except the vocabulary row which reports the primitive-level Damped Wave admissibility count and best error. Full setup details are in Appendix G.

| Axis | Ablation | Burgers | Diffusion | Damped Wave |
|---|---|---|---|---|
| Vocabulary | No atoms; 50,000 primitive-level CFG proposals | – | – | 133 admiss.; best $\approx 366\%$ |
| Search | HD-TLGP tree search over atom-level components | 2.04% | 33.34% | 423.30% |
| Full pipeline | SIGS latent search and refinement | $6.64 \times 10^{-14}$ | $7.16 \times 10^{-13}$ | $1.22 \times 10^{-13}$ |

**Latent topology.** After fixing the grammar, atom corpus, and latent-search paradigm, we isolate the topology-aware regulariser. The metric here is not PDE error but decoder efficiency away from the training set: how many decode attempts are needed to obtain 1000 valid off-training expressions. As shown in Table 7, TGVAE requires $3.56\%$ fewer attempts than GVAE, indicating that the regulariser improves the usable geometry of the latent manifold rather than changing the admissible symbolic family.

**Ansatz priors.** We test whether the admissible family must be minimal or narrowly matched. With the corpus and manifold fixed, we replace the one-term Burgers Ansatz in Table 2 by $u(x, t) = a_0 + \sum_{j=1}^{K} a_j \psi_j(x, t)$, where $K$ is the number of atom slots, and run $K = 1, 2, 3, 5$. SIGS reaches machine precision for $K \leq 3$ by suppressing redundant

*Table 7.* Ablation axis for latent topology. Both models use the same fixed corpus and decoder task; the metric is the number of decode attempts needed to obtain 1000 valid off-training expressions. Mean ± std across 10 disjoint sets of 15,000 latent samples.

| Model | Attempts | Effect |
|---|---|---|
| GVAE | $1486.2 \pm 19.5$ | baseline |
| TGVAE | $\mathbf{1433.2 \pm 27.3}$ | 3.56% fewer attempts |

terms and remains accurate at $K = 5$. The Poisson–Gauss experiment gives the complementary broad-family case: the masked superposition from Section (iii) yields 1–3% symbolic approximations despite no closed-form target. Together, these tests show that the admissible family must contain, or approximate, the target structure, but need not be minimal or narrowly matched.

**Grammar completeness.** The previous ablations vary atoms, search, topology, and Ansatz priors. We finally ablate the grammar's primitive set. For the Korteweg–de Vries equation, whose natural one-soliton solution is $2\,\text{sech}^2(x - 4t)$, we remove $\cosh$ and $\text{sech}$ from the grammar while $\tanh$ remains available. Using the Ansatz in Table 2, SIGS converges to $u \approx 2 - 2\tanh^2(x - 4t)$, equivalent to the true solution by the hyperbolic identity $\text{sech}^2(z) = 1 - \tanh^2(z)$, with $6.6 \times 10^{-6}$ relative $L^2$ error in 36s. Unlike the previous ablations, removing a primitive does *not* break SIGS: the latent search finds a residual-equivalent in-grammar representation. The atom corpus and Ansatz must contain or approximate the target, but specific primitives can be missing if their algebraic equivalents are reachable in the grammar.

## 4. Discussion and Conclusion

**Discussion.** This work advances solution discovery for PDEs by demonstrating that grammar-guided neuro-symbolic methods can reliably and efficiently recover analytical solutions. SIGS improves the state-of-the-art by orders of magnitude in accuracy and speed. Its success stems from two complementary design choices: (i) constructing a latent manifold of solution components, which enables smooth and efficient exploration of admissible expressions; and (ii) employing a hierarchical Ansatz+atom approach that reduces search complexity by structuring the solution space into manageable placeholders, later refined into concrete symbolic elements. This is in contrast to the baselines explored in this work, which do not address the combinatorial explosion inherent in symbolic solution discovery. HD-TLGP (Cao et al., 2024) transfers structures from one-dimensional solutions to higher dimensions, but still relies on stochastic recombination of primitives, which quickly becomes intractable as complexity grows. SSDE (Wei et al., 2024) instead uses reinforcement-learning to guide the construction of candidate solutions, but its flat search space remains prohibitively large without strong priors. As our experiments show, both methods degrade sharply when such priors are absent. In contrast, the hierarchical Ansatz+atom design of SIGS separates global structure from local symbolic details, making tractable what would otherwise be an unmanageable search. In this way, SIGS not only advances but redefines the state-of-the-art for solution discovery. Beyond these empirical gains, we view SIGS as part of a broader shift toward neuro-symbolic foundation models for PDEs. Current foundation approaches (Herde et al., 2024; Hao et al., 2024; Sun et al., 2024; Alkin et al., 2024; Shen et al., 2024) rely on extensive pretraining and often serve as black-box predictors for downstream tasks. In contrast, SIGS requires only a one-time pretraining step to construct its manifold, after which it transfers directly to new problems without retraining. Moreover, it produces analytical expressions that incorporate physical priors (e.g., eigenfunctions), yielding interpretable solutions rather than opaque approximations. This suggests that grammar-based neuro-symbolic models could complement or even provide an alternative to purely data-driven foundation models in scientific computing.

**Limitations.** Despite these contributions, SIGS faces two main limitations. First, scalability to complex engineering problems remains challenging. PDEs involving discontinuities, multiscale structure, or turbulence may require grammars enriched with special functions that cannot be easily decomposed into smaller atoms, or that produce long expressions which increase search complexity. Hybrid approaches that combine symbolic structures with numerical bases (e.g., POD-derived eigenfunctions, or Neural Operators) may provide a path forward, particularly for multiscale phenomena, as well as for problems with irregular geometries or boundary conditions. Second, although the KdV experiment shows robustness to moderate grammar mis-specification (missing a single primitive), the framework depends on the joint design of grammar, Ansatz, and latent space. A richer Ansatz can offset a simpler grammar, while a more expressive grammar requires larger latent spaces and more sophisticated optimization. Currently, the Ansatz still reflects human expert choices. This can be advantageous in domains with strong theoretical foundations (e.g., Burgers or Poisson equations), but limits applicability in less understood settings. A promising direction is to leverage large language models (e.g., Romera-Paredes et al., 2024) to automate Ansatz construction, learning general solution structures directly from governing equations.

**Conclusion.** In this work, we introduced the Symbolic Iterative Grammar Solver (SIGS), a grammar-guided neuro-symbolic framework for discovering analytical solutions to differential equations. By unifying classical compositional methods with modern latent-space optimization through the TGVAE, SIGS systematically explores the space of admissible solutions, enabling efficient search and refinement of closed-form expressions. Our approach achieves state-of-the-art performance on recent benchmarks, recovering exact solutions when available, and producing interpretable symbolic approximations for PDEs without known closed-form solutions. These results highlight the potential of grammar-based neuro-symbolic methods as a scalable and interpretable alternative to purely data-driven approaches, opening new directions for automated solution discovery in scientific computing.

## Acknowledgements

This work was made possible through the support of the ETH AI Center by PhD fellowships awarded to Orestis Oikonomou and Levi Lingsch.

## Impact Statement

This paper presents work whose goal is to advance the field of Machine Learning. There are many potential societal consequences of our work, none which we feel must be specifically highlighted here. By producing analytical formulas rather than opaque numerical approximations, and without requiring simulation data or per-problem retraining, SIGS makes formula discovery less dependent on rare human intuition and more available as a practical tool for science and engineering.

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

# Appendix Contents

The appendix is organized to make the implementation and evaluation choices traceable. A detailed roadmap for it is the following:

*Table 8.* Appendix roadmap.

| Appendix | Section | Purpose |
|---|---|---|
| A | Grammar, atoms, corpus construction | Fixed CFG, atom templates, corpus-time validity checks, corpus accounting, and post-hoc tags. |
| B | GVAE/TGVAE training | One-time manifold training, architecture, hyperparameters, hardware, and topology regularisation. |
| C | Solution search and refinement | Residual objective, Stage I search, Stage II coefficient refinement, and algorithms. |
| D | Benchmarks and Ansatze | PDE definitions, domains, grids, problem-specific assembly maps, and manufactured scalar references. |
| E | Baseline setups | HD-TLGP vanilla and atom-level setups, and SSDE primitive sets. |
| F | Detailed Results and Discovered Expressions | Prior-work parity, discovered expressions, scalar visualizations, and Poisson–Gauss reference validation. |
| G | Ablation and robustness | Mechanism ablations, admissible-prior checks, Burgers over-specification, and KdV primitive removal. |
| H | Coupled nonlinear systems | Shallow Water and Compressible Euler system definitions, Ansatze, expressions, and errors. |
| I | Classical tools, LLMs, runtime | Mathematica/LLM calibration, runtime table, LLM-use statement, and code availability. |

# A. Grammar, Atoms, and Corpus Construction

This appendix describes the shared symbolic vocabulary used by SIGS across all experiments. We first define the CFG representation, then describe how grammar-realizable atoms are generated from PDE-motivated templates and random CFG derivations, and finally specify the corpus-time checks, deduplication, and post-hoc tags used to organize the fixed atom corpus for downstream search.

## A.1. Formal Grammar and Expression Representation

The symbolic language is defined before any corpus is generated. We use a context-free grammar (CFG) $\mathcal{G} = \{\Phi, N, R, S\}$, where $\Phi$ is the terminal alphabet, $N$ is the nonterminal set, $R$ is the production-rule set, and $S \in N$ is the start symbol. The language $\mathcal{L}(\mathcal{G})$ contains all terminal strings derivable from $S$. Each expression can be represented equivalently as a string, a derivation tree, or a sequence of production-rule IDs. The GVAE operates on the production-rule sequence representation. The grammar contains 51 production rules; in the GVAE implementation the one-hot vocabulary has 52 channels because one padding rule symbol is included. The implementation uses the following rule schema, with compact digit ranges expanded into individual productions in the 51-rule CFG:

$$S \rightarrow S + T \mid S - T \mid S \times T \mid S/T \mid T \mid -T,$$
$$T \rightarrow (S) \mid (S)^2 \mid \sin(S) \mid \cos(S) \mid \exp(S) \mid \log(S) \mid \tanh(S) \mid \sqrt{S},$$
$$T \rightarrow \pi \mid x \mid y \mid t \mid x^2 \mid x^3 \mid y^2 \mid y^3 \mid D \mid D.D \mid -D \mid -D.D,$$
$$D \rightarrow 0 \mid 1 \mid 2 \mid 3 \mid 4 \mid 5 \mid 6 \mid 7 \mid 8 \mid 9 \mid D0 \mid D1 \mid D2 \mid D3 \mid D4 \mid D5 \mid D6 \mid D7 \mid D8 \mid D9 \mid e{-}1 \mid e{-}2 \mid e{-}3 \mid e{-}4.$$

**Terminology.** Primitives are the terminal-level tokens of the grammar, including variables, constants, and elementary function/operator symbols such as $x, t, +, \sin, \exp$. Atoms are complete grammar derivations that may contain many primitives and serve as component-level candidate functions. The corpus is the fixed finite collection of atoms used to train the GVAE/TGVAE. An Ansatz is the problem-level assembly map that combines selected atoms into a candidate solution, for example $u(x,t) = \sum_j \phi_j(x)\psi_j(t)$.

## A.2. Physics-Motivated Atom Templates

The physics-motivated atom stream uses the grammar above to instantiate reusable symbolic components. These grammar-realizable factors are later assembled by an Ansatz into candidate PDE solutions. The template stream includes separable spatial eigenmodes, temporal factors, characteristic or traveling phases, smooth transition layers, localized radial-basis

atoms, radial waves, and low-degree polynomial forms; the full corpus then augments these atoms with random CFG compositions over the same terminal vocabulary.

*Table 9.* Atom families generated by the fixed corpus pipeline. These are component-level functions in the searchable vocabulary; the Ansatz assembles selected atoms into full candidate solutions.

| Family | Examples | Purpose |
| --- | --- | --- |
| Spatial eigenmodes | $\sin(k\pi x/L)$, $\sin(k_x\pi x)\sin(k_y\pi y)$ | Dirichlet/separable spatial structure. |
| Temporal factors | $e^{-\lambda t}$, $\cos(\omega t)$, $\sin(\omega t)$ | Diffusion, wave, and damped-wave temporal behaviour. |
| Characteristic phases | $g(kx - \omega t)$ | Propagation along characteristic or traveling-wave coordinates. |
| Transition layers | $\tanh(ax + bt + c)$ | Smooth finite-width fronts and internal layers. |
| Localized radial-basis atoms | $\exp(-\alpha((x - x_0)^2 + (y - y_0)^2))$ | Local spatial resolution for source-driven elliptic/parabolic responses. |
| Radial waves | $\cos(k\sqrt{(x - x_0)^2 + (y - y_0)^2} - \omega t)e^{-\alpha t}$ | Outgoing or damped propagation in radial coordinates. |
| Random compositions | Grammar-generated combinations of the above primitives | Additional coverage beyond named PDE families. |

The operator-template stream first proposes parameterized expressions; every proposal then passes through corpus-time validity, canonicalization, and deduplication before entering the corpus. We keep two mathematically distinct sources separate. Tables 10 and 11 describe atoms obtained from the standard modal calculation for separable linear operators. Table 12 describes direct nonmodal atom families that encode common PDE representation mechanisms. The generator is run once, samples parameterized instances from these families, and expresses each sampled instance as a derivation in $\mathcal{G}$.

*Table 10.* Laplacian eigenfamilies used by the operator-template stream on box domains, with $A = -\Delta$, $A\phi_{\mathbf{k}} = \mu_{\mathbf{k}}\phi_{\mathbf{k}}$, and $\Omega = \prod_d[0, L_d]$.

| Boundary type | Spatial factor $\phi_{\mathbf{k}}$ | Eigenvalue $\mu_{\mathbf{k}}$ | Index set |
| --- | --- | --- | --- |
| Dirichlet | $\prod_d \sin(k_d\pi x_d/L_d)$ | $\pi^2\sum_d k_d^2/L_d^2$ | $k_d \in \mathbb{N}$ |
| Neumann | $\prod_d \cos(k_d\pi x_d/L_d)$ | $\pi^2\sum_d k_d^2/L_d^2$ | $k_d \in \mathbb{N}_0$ |
| Periodic | $\prod_d\{\sin, \cos\}(2\pi k_d x_d/L_d)$ | $4\pi^2\sum_d k_d^2/L_d^2$ | $k_d \in \mathbb{Z}$ |

*Table 11.* Modal temporal factors used by the operator-template stream. For separable linear operators, projecting onto $A\phi_{\mathbf{k}} = \mu_{\mathbf{k}}\phi_{\mathbf{k}}$ gives the scalar ODE shown below. These rows are derived modal factors shared by the corpus generator.

| Family | Modal equation | Generated temporal factor |
| --- | --- | --- |
| Heat / diffusion | $T_{\mathbf{k}}' + \kappa\mu_{\mathbf{k}}T_{\mathbf{k}} = 0$ | $\exp(-\kappa\mu_{\mathbf{k}}t)$ |
| Wave | $T_{\mathbf{k}}'' + c^2\mu_{\mathbf{k}}T_{\mathbf{k}} = 0$ | $\cos(c\sqrt{\mu_{\mathbf{k}}}t), \sin(c\sqrt{\mu_{\mathbf{k}}}t)$ |
| Damped wave / telegraph | $T_{\mathbf{k}}'' + 2\gamma T_{\mathbf{k}}' + c^2\mu_{\mathbf{k}}T_{\mathbf{k}} = 0$ | $e^{-\gamma t}\cos(\omega_{\mathbf{k}}t), e^{-\gamma t}\sin(\omega_{\mathbf{k}}t)$ |
| Biharmonic / plate | $T_{\mathbf{k}}' + \kappa\mu_{\mathbf{k}}^2 T_{\mathbf{k}} = 0$ or second-order variant | $\exp(-\kappa\mu_{\mathbf{k}}^2 t)$, damped sinusoidal variants |
| Reaction–diffusion linear part | $T_{\mathbf{k}}' + (\kappa\mu_{\mathbf{k}} - \rho)T_{\mathbf{k}} = 0$ | $\exp(-(\kappa\mu_{\mathbf{k}} - \rho)t)$ |

*Table 12.* Direct nonmodal atom families. These atoms are sampled directly as grammar-realizable expressions and then subjected to the same validity, canonicalization, and deduplication checks as all other corpus entries. They encode standard PDE representation mechanisms: characteristic propagation, smooth transition layers, local radial-basis resolution, and radial propagation.

| Atom family | Sampled phase or spatial form | Generated expression family |
| --- | --- | --- |
| Characteristic / traveling phase | $kx - \omega t + c$ | $g(kx - \omega t + c)$, $g \in \{\sin, \cos, \tanh, \exp\}$ when valid |
| Viscous transition layer | $(x - st - x_0)/\ell$ with sampled speed, center, and width | $\tanh(ax + bt + c)$ |
| Localized radial-basis atom | $(x - x_0)^2 + (y - y_0)^2$ with sampled center and width | $\exp(-\alpha((x - x_0)^2 + (y - y_0)^2))$ |
| Radial wave / envelope | $\sqrt{(x - x_0)^2 + (y - y_0)^2}$, optionally with $t$ | $\cos(kr - \omega t + c)$, $e^{-\alpha t}\cos(kr - \omega t + c)$ |

The direct families have standard PDE interpretations. Characteristic phases represent profiles that are transported along level sets, as in $u_t + cu_x = 0$ where $u(x, t) = g(x - ct)$. Smooth transition layers represent finite-width fronts and internal layers that arise in viscous conservation laws and singularly perturbed advection–diffusion problems. Localized radial-basis atoms provide local spatial resolution: for elliptic problems, Green's representation writes $u(x) = \int_\Omega G(x, \xi) f(\xi) \, d\xi$, so localized forcing is naturally approximated by local smooth response components. Radial wave atoms encode dependence on distance from a source or center, a common coordinate for outgoing waves in homogeneous media. These are mechanism-level atoms; the downstream residual determines whether any of them is selected for a given PDE.

For separable constant-coefficient operators on simple domains, operator-motivated generation uses standard eigenfamilies. For example, on a rectangular domain with homogeneous Dirichlet conditions,

$$\phi_{\mathbf{k}}(\mathbf{x}) = \prod_{d=1}^{D} \sin(k_d \pi x_d / L_d), \qquad \mu_{\mathbf{k}} = \pi^2 \sum_{d=1}^{D} k_d^2 / L_d^2. \tag{3}$$

These spatial modes can be paired with temporal factors such as $e^{-\kappa \mu_{\mathbf{k}} t}$ for diffusion or $\cos(c\sqrt{\mu_{\mathbf{k}}} t)$ for waves. The same corpus also includes the direct nonmodal atoms in Table 12, giving the searchable vocabulary both separated and nonseparated components.

Initial template constants are sampled during one-time corpus construction from fixed ranges. Mode indices are drawn from bounded low-order integer sets; amplitudes, speeds, centers, widths, damping rates, and diffusion/wave parameters are drawn from fixed bounded ranges chosen to avoid numerical overflow on the validation boxes. Numeric literals are rounded to three decimals for most atoms and to six decimals for oscillatory wave parameters; symbolic terminals such as $\pi$ remain exact grammar symbols. After Stage I selects a symbolic structure, Stage II may refine remaining numerical literals by residual minimization. Boundary-compatible envelopes, e.g. $\sin(\pi x / L_x) \sin(\pi y / L_y)$, are included as ordinary atoms for homogeneous Dirichlet spatial structure, while cosine modes cover Neumann-type structure.

*Table 13.* Sampling ranges used by the operator-template stream. These ranges are fixed during one-time corpus construction and shared by the downstream benchmarks.

| Template family | Sampled parameters |
| --- | --- |
| Wave / oscillatory modes | Low-order mode indices; wave speed $c \in [0.1, 0.8]$; amplitude mantissa $m \in [5, 9]$ with scientific-notation scaling for numerical stability. |
| Diffusion / heat modes | Odd modes $2n + 1$ with $n \in \{0, 1, 2\}$; initial magnitude $M_0 \in [1, 3]$; domain length $L \in [0.1, 1.5]$; diffusivity $D \in [0.01, 1]$. |
| Viscous transition layers | Left state $u_L \in [1, 3]$; right state $u_R \in [-1, 1]$; speed $s \in [0.1, 2]$; center $x_0 \in [-1, 1]$; viscosity $\nu \in [0.01, 1]$. |
| Localized radial-basis atoms | Centers sampled on the corresponding spatial validation box; widths/decays sampled from bounded numerically stable ranges. |
| Radial damped waves / envelopes | Amplitude $h \in [0.01, 0.5]$; envelope width $w \in [0.3, 1.0]$; wave number $k \in [0.5, 4.0]$; phase velocity $c \in [0.1, 1.0]$; decay $a \in [0.02, 0.8]$; center $(x_0, y_0) \in [-6, 6]^2$. |
| Random CFG expansions | Binary/unary/terminal expansion probabilities $0.6/0.3/0.1$, followed by the same validity, canonicalization, and deduplication checks. |

### A.3. Corpus-Time Validity, Canonicalization, and Deduplication

Before describing corpus accounting, we specify the acceptance rule applied to every proposed expression from every stream. Each proposal is first represented as a CFG derivation sequence and then checked for numerical and representational suitability. These checks belong to corpus construction; during SIGS search, tag-admissible decoded candidates are assembled and scored directly by the residual.

The corpus-time validity checks remove expressions that are syntactically complete but unsuitable for numerical residual evaluation:

- variable-presence mismatches, e.g. one-variable streams must contain their declared variable and multivariable streams must contain the declared spatial or spatiotemporal variables;

- domain violations such as logarithms of nonpositive expressions or square roots of negative expressions on fixed validation boxes used during corpus construction;

- divisions whose denominator is numerically near zero on the validation grid;

- purely constant expressions when a variable-dependent atom is required;

- numerically unstable exponentials or constants outside the permitted range;

- expressions exceeding the production-sequence length budget.

Accepted expressions are then canonicalized before insertion into the corpus. Canonicalization includes simple algebraic normalization such as rewriting reciprocal notation, standardizing square-root forms, and simplifying numeric products when this is unambiguous. A global hash table keeps one canonical copy of each expression, so duplicate expressions proposed by different streams or parameter draws are counted once. These steps define the stable atom corpus used for GVAE/TGVAE training. During downstream search, decoded candidates are assembled and scored by the PDE residual after the component tags required by the Ansatz are enforced.

### A.4. Corpus Generation Pipeline

The corpus is generated once from the fixed CFG and then reused for all downstream experiments. The generator randomly mixes two sources of expressions. First, the template stream samples parameters in the operator-motivated expression families above and retains only instances that are realizable in the fixed CFG, representing each accepted instance as a production-rule derivation sequence. Second, the random stream samples derivations directly from the same CFG under fixed depth, length, and variable-set limits, producing one-variable and multivariable compositions over variables such as $(x)$, $(x, y)$, $(x, t)$, and $(x, y, t)$. We do not tune a template/random ratio or balance the corpus per benchmark; one corpus is generated, validated, canonicalized, deduplicated, and then used for all experiments.

For transparency, we report the resulting corpus accounting after this one-time generation. The random CFG expressions contribute approximately 15,000 candidates before the final global deduplication pass: about 5,000 one-variable expressions and 10,000 multivariable expressions. The remaining retained expressions come from the operator-motivated template source. Since validation and deduplication are global, provenance is reported as approximate retained contribution rather than as controlled stream quotas. After validation, canonicalization, and deduplication, the final corpus contains 23,682 expression sequences. Appendix B gives the production-rule encoding, train/validation/test split, architecture, optimizer, and training schedules used to train the GVAE/TGVAE on this corpus.

*Table 14.* Corpus-construction accounting for the one-time generated corpus. Counts are approximate retained contributions before the final global deduplication pass; duplicate expressions can be proposed by both sources, so the final downstream corpus is the 23,682-expression set.

| Source | Accounting | Generated content |
|---|---|---|
| Operator-motivated templates | Remaining retained expressions after random CFG expressions and global deduplication | Eigenmodes, time factors, characteristic phases, transition layers, localized radial-basis atoms, radial waves, boundary envelopes, and low-degree polynomials with sampled constants. |
| Random CFG compositions | $\approx$ 15,000 expressions total | Generic grammar expansions: $\approx$ 5,000 one-variable expressions and $\approx$ 10,000 multivariable expressions over $(x, y)$, $(x, t)$, or $(x, y, t)$. |
| Final corpus | 23,682 sequences | Valid, canonicalized, deduplicated production-rule sequences used once for GVAE/TGVAE training. |

#### A.4.1. Post-Hoc Tags

Variable tags are computed from the decoded parse tree $\mathcal{D}(\mathcal{E}(w))$ after the trained encoder–decoder round-trip, so tags reflect what the decoder actually produces during search rather than the original corpus atom. Typical tags include $x$-only, $t$-only, $(x, t)$, $(x, y)$, or $(x, y, t)$. The tags are admissibility labels used to populate component-specific libraries. For example, a temporal slot in a separable Ansatz uses atoms containing $t$, while a spatial slot uses atoms containing the corresponding spatial variables. The grammar generates the expressions; the tags organize generated atoms for the assembly map.

## B. GVAE/TGVAE Training

This appendix gives the one-time training details for the latent manifold used by all downstream experiments. The corpus from Appendix A is represented as production-rule sequences, trained with a grammar-masked GVAE objective, and optionally regularised with the topology-aware losses used by TGVAE. No downstream PDE residuals, boundary conditions, benchmark labels, or Ansätze enter this training step; after training, the encoder/decoder are frozen and reused for all target equations.

## B.1. Model and Data

We train a Grammar Variational Autoencoder (GVAE) on the fixed atom corpus. Inputs are one-hot production-rule sequences with shape $N \times C \times L = 23682 \times 52 \times 72$, where $C = 52$ includes the 51 grammar productions plus one padding rule symbol and $L = 72$ is the maximum sequence length. The encoder uses three 1D convolutional layers followed by two heads for $\mu$ and $\log \sigma^2$ in a 32-dimensional latent space. The decoder is positional: it lifts $z \in \mathbb{R}^{32}$ to a hidden state, runs a GRU over sequence positions, and outputs production logits at each position under the grammar mask. Table 15 gives the architecture used for both GVAE and TGVAE.

Training is standard teacher-forced sequence VAE training over CFG production rules. For each minibatch, a derivation sequence is encoded into $q_\phi(z \mid w) = \mathcal{N}(\mu_\phi(w), \operatorname{diag} \sigma_\phi^2(w))$, a latent sample is drawn by the reparameterization trick, and the decoder predicts all 72 production-rule positions under the grammar mask. The reconstruction loss is cross-entropy against the target rule sequence; the KL term is warmed up; TGVAE adds the geometric terms in Appendix B.3. We monitor validation ELBO, sequence-exact accuracy, grammar-valid decode rate, and the individual loss components, and stop early after 10 epochs without validation-ELBO improvement.

*Table 15.* GVAE/TGVAE architecture. All sequence lengths refer to production-rule sequences; the TGVAE uses the same encoder/decoder and adds the geometric losses in Section B.3.

| Block | Layer | Dimensions / settings |
|---|---|---|
| Input | One-hot sequence | $C = 52, L = 72$ |
| Encoder | Conv1D + ELU | $52 \to 64$, kernel 2, length $72 \to 71$ |
| Encoder | Conv1D + ELU | $64 \to 128$, kernel 3, length $71 \to 69$ |
| Encoder | Conv1D + ELU | $128 \to 256$, kernel 4, length $69 \to 66$ |
| Encoder | Dense + heads | $256 \times 66 \to 256$, then $256 \to 32$ for $\mu$ and $\log \sigma^2$ |
| Decoder | Dense + GRU | $32 \to 512$, one GRU layer with hidden size 512 |
| Decoder | Time-distributed output | $512 \to 52$ logits at each of 72 positions |
| Model size | Trainable parameters | approximately 6.1M parameters |

*Table 16.* Corpus split used for GVAE/TGVAE training.

| Split | Number of sequences |
|---|---|
| Train | 16,578 |
| Validation | 4,736 |
| Test | 2,368 |

*Table 17.* GVAE/TGVAE training settings.

| Item | Value |
|---|---|
| Hardware | single NVIDIA GeForce RTX 5080 Laptop GPU, 16 GB VRAM |
| Software | Python 3.10.18; PyTorch 2.7.1+cu128; Lightning 2.5.2; CUDA 12.8; cuDNN 90800 |
| Latent dimension | 32 |
| Optimizer | AdamW, learning rate $3 \times 10^{-4}$, weight decay $10^{-5}$ |
| Batch size | 64 for train and validation, 4 dataloader workers |
| Precision | mixed precision with global gradient clipping at 1.0 |
| Scheduler | ReduceLROnPlateau, factor 0.2, patience 5, monitoring validation ELBO |
| KL schedule | linear warmup to 1.0 over 7000 steps, initial weight 0.01 |
| Topological activation | enabled after validation sequence-exact accuracy reaches 20%, then ramped over 5 epochs |
| Topological schedule | computed sparsely, every 50 training steps and every 12 validation batches |
| Topological weights | $w_{\text{Hull}} = 0.8$, $w_{\text{ph}} = 0.8$, $w_{\text{smooth}} = 10^{-4}$ |
| Persistent homology settings | Rips complex on CPU, max 24 points, max dimension 1, scales $\{0.10, 0.50\}$ |
| Hull settings | 256 fixed directions in the 32-dimensional latent space |
| Train/validation budget | up to 200 epochs with early stopping patience 10 |

## B.2. GVAE Objective

For a production-rule sequence $w = (r_1, \ldots, r_L)$, the encoder defines

$$q_\phi(z \mid w) = \mathcal{N}\left(\mu_\phi(w), \operatorname{diag} \sigma_\phi^2(w)\right), \qquad p(z) = \mathcal{N}(0, I).$$

The decoder predicts a distribution over the 52 rule channels at each sequence position. At each position, logits for CFG-inadmissible productions are masked before the softmax, so reconstruction is evaluated only over grammar-valid next-rule choices. The reconstruction term is

$$\mathcal{L}_{\text{recon}}(w, z) = -\sum_{\ell=1}^{L} \log p_\theta(r_\ell \mid z, \ell, \mathcal{G}), \tag{4}$$

where the padding rule is treated as the target after the derivation terminates. The vanilla GVAE objective used for the non-topological baseline is

$$\mathcal{L}_{\text{GVAE}}(w) = \mathbb{E}_{z \sim q_\phi(z|w)} \left[ \mathcal{L}_{\text{recon}}(w, z) \right] + \beta(t) \operatorname{KL}(q_\phi(z \mid w) \,\|\, p(z)), \tag{5}$$

with the closed-form diagonal-Gaussian KL

$$\operatorname{KL}(q_\phi(z \mid w) \,\|\, p(z)) = \frac{1}{2} \sum_{j=1}^{32} \left( \mu_j^2 + \sigma_j^2 - \log \sigma_j^2 - 1 \right). \tag{6}$$

The KL coefficient $\beta(t)$ is linearly warmed up according to Table 17. We report validation ELBO using the same reconstruction and KL terms, and use sequence-exact accuracy and grammar-valid decode rate as auxiliary diagnostics.

## B.3. Geometry regularisation

The TGVAE augments the GVAE objective with three geometric terms. A convex-hull penalty discourages latent points from drifting outside an empirical enclosure of the training reservoir; a persistent-homology penalty suppresses small holes and disconnected micro-clusters at the working resolution; and a decoder smoothness penalty discourages sharp local curvature in the decode map. The full TGVAE objective is

$$\mathcal{L}_{\text{TGVAE}} = \mathcal{L}_{\text{GVAE}} + \gamma(e)\left(0.8\mathcal{L}_{\text{Hull}} + 0.8\mathcal{L}_{\text{ph}} + 10^{-4}\mathcal{L}_{\text{smooth}}\right). \tag{7}$$

Here $\gamma(e)$ ramps the topology terms after the validation sequence-exact accuracy reaches 20%.
Let $z_i = \mu_\phi(w_i)$ be encoder means in the current minibatch, $\mathcal{Z}_R$ a fixed-size reservoir of previous latent codes, and $\{h_j\}_{j=1}^{K} \subset \mathbb{S}^{31}$ the $K = 256$ fixed hull directions. With projection bounds $a_j^- = \min_{z \in \mathcal{Z}_R} h_j^\top z$ and $a_j^+ = \max_{z \in \mathcal{Z}_R} h_j^\top z$,

$$\mathcal{L}_{\text{Hull}} = \frac{1}{BK} \sum_{i,j} \left[ \operatorname{ReLU}(h_j^\top z_i - a_j^+)^2 + \operatorname{ReLU}(a_j^- - h_j^\top z_i)^2 \right]. \tag{8}$$

For $\mathcal{L}_{\text{ph}}$, we compute Vietoris–Rips persistent homology on at most 24 latent points drawn from the current minibatch and latent reservoir. Let $V_k(P)$ denote the $k$-dimensional persistence diagram of the sampled point cloud $P$. At working radius $r = \sqrt{2}\delta$, define the clamped lifetime

$$\ell_r(b, d) = \max\{0, \min(d, r) - \min(b, r)\}. \tag{9}$$

The persistent-homology penalty is

$$\mathcal{L}_{\text{ph}}(P) = \sum_{(b,d) \in V_1(P)} \ell_r(b, d)^2 \;+\; a_0 \sum_{(b,d) \in V_0(P)} \ell_r(b, d)^2, \tag{10}$$

with $a_0$ fixed. This term suppresses holes and disconnected micro-clusters at the working resolution. For decoder smoothness, with unit random probes $v$ and decoder-logit map $D_\theta$,

$$\mathcal{L}_{\text{smooth}} = \mathbb{E}_{z,v} \left[ \left\| \frac{D_\theta(z + \epsilon v) - 2D_\theta(z) + D_\theta(z - \epsilon v)}{\epsilon^2} \right\|_2^2 \right], \tag{11}$$

using $\epsilon = 10^{-3}$ and one probe per sampled latent point. The GVAE/TGVAE ablation uses a race-to-valid sampling protocol. For each model, we sample latent points away from the training encodings by a shared Mahalanobis-distance exclusion rule, decode until 1000 valid grammar strings are obtained, and count the number of decode attempts required. Given latent point $z$, training encoder means $\{\mu_i\}_{i=1}^N$, and covariance $\Sigma$, the distance is $d_M(z, \mu_i) = \sqrt{(z - \mu_i)^\top \Sigma^{-1}(z - \mu_i)}$; samples are accepted when $\min_i d_M(z, \mu_i) \geq 0.8$ for both models, which evaluates off-training decodes under the same exclusion rule. We sample 15,000 admissible latent vectors and split them into ten disjoint sets, providing each set to both models in turn. Under this protocol, the vanilla GVAE required $1486.2 \pm 19.5$ decode attempts, while TGVAE required $1433.2 \pm 27.3$, a $3.56\% \pm 1.81$ reduction. This ablation measures the sampling-efficiency contribution of the geometric regulariser.

## C. Solution Search and Coefficient Refinement

After GVAE/TGVAE training, the manifold is fixed. For each target PDE, SIGS uses the practitioner-specified Ansatz to select tag-admissible atom libraries, searches the trained latent space for symbolic structures with low residual, and then refines the numerical literals in the selected expression. The trained GVAE/TGVAE is reused during the downstream solve.

### C.1. Downstream Search Objective

For a decoded and assembled candidate $u$, SIGS minimizes the discretized residual

$$R(u) = \frac{1}{|\mathcal{M}|} \sum_{\mathbf{x} \in \mathcal{M}} \frac{1}{|\mathcal{T}|} \sum_{t \in \mathcal{T}} (\mathcal{S}[u](\mathbf{x}, t))^2 \tag{12}$$

$$+ \beta_1 \frac{1}{|\mathcal{M}_{IC}|} \sum_{\mathbf{x} \in \mathcal{M}_{IC}} (u(\mathbf{x}, 0) - u_0(\mathbf{x}))^2 \tag{13}$$

$$+ \beta_2 \frac{1}{|\mathcal{M}_{BC}|} \sum_{\mathbf{x} \in \mathcal{M}_{BC}} \frac{1}{|\mathcal{T}|} \sum_{t \in \mathcal{T}} (\mathbb{B}[u](\mathbf{x}, t) - g(\mathbf{x}, t))^2. \tag{14}$$

All candidate expressions are evaluated through this residual after decoding and assembly.

*Table 18.* Downstream SIGS search settings. Exact per-problem scripts and random seeds will be released with the code.

| Stage | Setting |
|---|---|
| Stage I-A initial search | Per slot, draw one atom per latent cluster. Enumerate cross-slot cluster tuples and assemble via $A$; score each by residual in parallel where possible. The lowest-residual tuple identifies the winning cluster per slot. Approximately 1000 cross-slot combinations are evaluated per round. |
| Stage I-B focused refinement | Iteratively restrict to the current incumbent component clusters, repartition those clusters into subclusters, and continue sampling/decoding assembled candidates until the structural residual reaches the problem-dependent threshold or the budget is exhausted. We use structural thresholds on the order of $10^{-2}$–$10^{-3}$. |
| Stage II coefficient refinement | Refine numerical literals only after a symbolic structure is selected. Optimization is implemented in float64 JAX with Adam (Optax) under exponential learning-rate decay and JIT, all multi-starts optimized in parallel via `vmap`. Across problems we use 100–500 multi-starts ($N_{\text{start}}$) and noise scale $\eta \in [0.1, 0.5]$ on the initial-coefficient draw, with Adam's default initial learning rate ($\eta_0 = 10^{-3}$); per-problem values are listed in the released configuration files. Optimization runs until $\sqrt{R(u)} < \varepsilon_{\text{tol}}$ or the per-problem iteration budget is reached. |
| Evaluation metric | Known-solution and coupled-system errors are reported as relative $L^2$ errors on the evaluation grids; PG errors use the FEniCS projection protocol in Appendix F.4. |

### C.2. Solution-Search Algorithms

The algorithms use the following notation. $\mathcal{L}$ is the fixed atom corpus, $c \in \{1, \ldots, L\}$ indexes Ansatz component slots, $\mathcal{C}_c$ is the tag/admissibility requirement for slot $c$, $A$ assembles component atoms into a full candidate expression, $\mathcal{I}$ interprets a grammar string as a function, and $R$ is the discretized PDE/condition residual. A latent vector for component $c$ is denoted $z^{(c)}$, and a cluster tuple $k = (k^{(1)}, \ldots, k^{(L)})$ records which component clusters are being searched.

---

**Algorithm 2** SIGS overview

---

**Require:** Grammar-generated atom corpus $\mathcal{L}$, trained encoder/decoder $(\mathcal{E}, \mathcal{D})$, interpretation map $\mathcal{I}$, Ansatz assembly map $A$, component tag requirements $\{\mathcal{C}_c\}_{c=1}^L$, residual $R$, budgets $(M, T_{\max})$, coefficient starts $N_{\text{start}}$, thresholds $(\varepsilon_{\text{struct}}, \varepsilon_{\text{tol}})$
**Ensure:** Refined symbolic expression $w^\star(p^\star)$
1: Build component libraries $\mathcal{L}^{(c)} = \{w \in \mathcal{L} : w \text{ satisfies } \mathcal{C}_c\}$
2: $(w^\star, z^\star, k^\star, r^\star) \leftarrow \text{INITIALLIBRARYSEARCH}(\{\mathcal{L}^{(c)}\}, \mathcal{E}, \mathcal{D}, A, \mathcal{I}, R, M)$
3: $(w^\star, z^\star, k^\star, r^\star) \leftarrow \text{SUBCLUSTERREFINEMENT}(w^\star, z^\star, k^\star, r^\star, \mathcal{D}, A, \mathcal{I}, R, T_{\max}, \varepsilon_{\text{struct}})$
4: $p^\star \leftarrow \text{COEFFICIENTREFINEMENT}(w^\star, \mathcal{I}, R, N_{\text{start}}, \varepsilon_{\text{tol}})$
5: **return** $w^\star(p^\star)$

---

**Algorithm 3** Stage I-A: initial library search

---

**Require:** Component libraries $\{\mathcal{L}^{(c)}\}_{c=1}^L$, encoder/decoder $(\mathcal{E}, \mathcal{D})$, assembly map $A$, interpretation map $\mathcal{I}$, residual $R$, draw budget $M$
**Ensure:** Best candidate $(w^\star, z^\star, k^\star, r^\star)$
1: Encode all corpus atoms $Z = \{\mathcal{E}(w) : w \in \mathcal{L}\}$; for each $z \in Z$, decode and compute the tag $\tau(\mathcal{D}(z))$
2: **for** each component $c$ **do**
3:     Form the tag-admissible latent subset $Z^{(c)} = \{z \in Z : \tau(\mathcal{D}(z)) \in \mathcal{T}^{(c)}\}$ and cluster into $K_c$ clusters via k-means
4: **end for**
5: Initialize $r^\star \leftarrow \infty$
6: **for** each component $c$ and each cluster $k \in \{1, \dots, K_c\}$ **do**
7:     Draw $z_k^{(c)}$ from cluster $k$, decode $w_k^{(c)} = \mathcal{D}(z_k^{(c)})$; skip if malformed
8: **end for**
9: **for** each cluster tuple $k = (k^{(1)}, \dots, k^{(L)}) \in \prod_c \{1, \dots, K_c\}$ **do**
10:     Assemble $w_k = A(w_{k^{(1)}}^{(1)}, \dots, w_{k^{(L)}}^{(L)})$ and score $r_k = R(\mathcal{I}(w_k))$
11:     **if** $r_k < r^\star$ **then**
12:         $(w^\star, k^\star, r^\star) \leftarrow (w_k, k, r_k)$
13:     **end if**
14: **end for**
15: **return** $(w^\star, z^\star, k^\star, r^\star)$

---

**Algorithm 5** Stage II: coefficient refinement

---

**Require:** Best symbolic structure $w^\star$, interpretation map $\mathcal{I}$, residual $R$, tolerance $\varepsilon_{\text{tol}}$, starts $N_{\text{start}}$, noise scale $\eta$
**Ensure:** Optimized coefficients $p^\star$
1: Parse numeric literals in $w^\star$ into $\bar{p}$; protect symbolic constants such as $\pi$, $e$, and integer exponents
2: **for** $s = 1$ to $N_{\text{start}}$ **do**
3:     Initialize $p^{(0,s)} \sim \mathcal{N}(\bar{p}, \text{diag}((\eta|\bar{p}|)^2))$
4: **end for**
5: Optimize all starts in float64 JAX with Adam and automatic differentiation through the PDE residual
6: Keep $p^\star = \arg\min_s R(\mathcal{I}(w^\star(p^{(s)})))$
7: **return** $p^\star$

---

# D. Benchmarks and Ansatze

This appendix fixes the experimental problem statements and the assembly families used by SIGS. The PDE table gives the residuals, domains, grids, and unknown fields, while the Ansatz table specifies only the high-level composition rule; the atoms that fill those slots always come from the fixed corpus in Appendix A.

## D.1. Problem Definitions

All benchmarks are written in the same residual notation used by the search objective. For scalar problems, SIGS evaluates $\mathcal{S}[u] = 0$ together with the prescribed boundary/initial conditions. For manufactured systems, the forcing is moved to the

---

**Algorithm 4** Stage I-B: focused subcluster refinement

---

**Require:** Incumbent $(w^\star, z^\star, k^\star, r^\star)$, component embeddings and cluster assignments from Stage I-A, decoder $\mathcal{D}$, assembly map $A$, interpretation map $\mathcal{I}$, residual $R$, budget $T_{\max}$, threshold $\varepsilon_{\text{struct}}$
**Ensure:** Updated $(w^\star, z^\star, k^\star, r^\star)$
 1: $t \leftarrow 0$
 2: **while** $r^\star > \varepsilon_{\text{struct}}$ and $t < T_{\max}$ **do**
 3:     For each component $c$, restrict to winning cluster $k^{\star(c)}$ and repartition into $K'_c$ sub-clusters via k-means
 4:     **for** each component $c$ and each sub-cluster $k' \in \{1, \ldots, K'_c\}$ **do**
 5:         Draw/decode $w_{k'}^{(c)}$; add convex-interpolation jitter if sub-cluster too small; skip if malformed
 6:     **end for**
 7:     **for** each sub-cluster tuple $k' = (k'^{(1)}, \ldots, k'^{(L)})$ **do**
 8:         Assemble $w_{k'} = A(w_{k'^{(1)}}^{(1)}, \ldots, w_{k'^{(L)}}^{(L)})$; compute $r_{k'} = R(\mathcal{I}(w_{k'}))$
 9:         **if** $r_{k'} < r^\star$ **then**
10:             Update incumbent $(w^\star, k^\star, r^\star) \leftarrow (w_{k'}, k', r_{k'})$; winning sub-clusters become next-round clusters
11:         **end if**
12:     **end for**
13:     $t \leftarrow t + 1$
14: **end while**
15: **return** $(w^\star, z^\star, k^\star, r^\star)$

---

left-hand side, so the evaluated residual is $\mathcal{S}[U] - f = 0$. The condition terms in $R(u)$ use the same domains and grids listed below.

*Table 19.* Benchmark PDEs in residual form. Dimension notation $n+m$D denotes $n$ spatial and $m$ temporal dimensions.

| Problem | $\mathcal{S}[\cdot]$ evaluated in the residual | Unknown | Domain | Grid |
|---|---|---|---|---|
| Burgers | $u_t + uu_x - \nu u_{xx} = 0$ | $u(x, t)$ | $[-5, 5] \times [0, 2]$ | $128^2$ |
| Diffusion | $u_t - \kappa u_{xx} = 0$ | $u(x, t)$ | $[0, 1.397] \times [0, 1]$ | $128^2$ |
| Damped wave | $u_{tt} + u_t - c^2(u_{xx} + u_{yy}) = 0$ | $u(x, y, t)$ | $[-8, 8]^2 \times [0, 4]$ | $32^3$ |
| Poisson–Gauss | $-\Delta u - f = 0,\ u\vert_{\partial\Omega} = 0$ | $u(x, y)$ | $[0, 1]^2$ | $100^2$ |
| KdV | $u_t + 6uu_x + u_{xxx} = 0$ | $u(x, t)$ | $[-10, 10] \times [0, 1]$ | $128^2$ |
| Shallow Water | $\mathcal{S}_{\text{SWE}}[U] - f = 0$, Eq. (28) | $U = (\rho, u, v)$ | $[-10, 10]^2 \times [0, 5]$ | $64^2 \times 32$ |
| Compressible Euler | $\mathcal{S}_{\text{CE}}[U] - f = 0$, Eq. (29) | $U = (\rho, u, v, p)$ | $[0, 1]^2$ | $64^2$ |

## D.2. Problem-Specific Ansatze

*Table 20.* Problem-specific Ansatze. These define the assembly family; the atoms that fill the slots come from the fixed corpus.

| Problem | Ansatz | Atom slots |
|---|---|---|
| Burgers | $u(x, t) = a_0 + a_1\phi(x, t)$ | one spatiotemporal shock/transport atom plus coefficients |
| Diffusion | $u(x, t) = \sum_{j=1}^3 a_j\phi_j(x)\psi_j(t)$ | three spatial atoms and three temporal atoms |
| Damped wave | $u(x, y, t) = \phi(x, y, t)\psi(t)$ | radial or oscillatory spatiotemporal atom, temporal decay atom |
| Poisson–Gauss | $u(x, y) = \sin(\pi x)\sin(\pi y)\sum_{j=1}^K a_j\phi_j(x, y)$ | generic superposition of spatial atoms; $K$ increases with the number of sources |
| KdV | $u(x, t) = \sum_{k=0}^2 a_k\phi(x, t)^k$ | one spatiotemporal atom and a quadratic polynomial assembly |
| SWE | $\rho = \phi_1\phi_2\phi_3,\ u = \rho\,\psi_x,\ v = \rho\,\psi_y$ | coupled density and velocity atoms |
| Compressible Euler | $\rho = \exp(\sum_{i=1}^6 f_i),\ u = \sum_{i=1}^6 g_i,\ v = \sum_{i=1}^6 h_i,\ p = \exp(\sum_{i=1}^6 k_i)$ | 24 spatial atom slots across four fields |

For the representable scalar PDEs, manufactured solutions are used so that error can be measured directly. For Poisson–Gauss, no closed-form reference is used; results are evaluated against a validated FEniCS numerical reference. For coupled systems, manufactured forcing terms define the PDE residuals, and SIGS searches jointly over all fields.

*Table 21.* Manufactured scalar solutions used for exact-error evaluation. Boundary and initial conditions are taken from these expressions.

| Problem | Reference solution $u^\star$ | Constants |
|---|---|---|
| Burgers | $0.86 + 0.6\tanh(25.8t - 30.0x + 9.9)$ | $\nu = 0.01$ |
| Diffusion | $A\left[\sin\left(\frac{\pi x}{L}\right)e^{-\pi^2\kappa t/L^2} - \sin\left(\frac{3\pi x}{L}\right)e^{-9\pi^2\kappa t/L^2} + \sin\left(\frac{5\pi x}{L}\right)e^{-25\pi^2\kappa t/L^2}\right]$ | $A = 3.974$, $L = 1.397$, $\kappa = 0.697$ |
| Damped wave | $e^{-\alpha t}\cos(\omega t - KR(x,y))$, $R(x,y) = \sqrt{(hx+1)^2 + (hy-1)^2}$ | $h = 0.2$, $K = 2.5$, $\omega = 0.4$, $\alpha = 0.45$ |

## E. Baseline Setups

This appendix records what information is supplied to each baseline and how the comparisons should be read. We give two HD-TLGP setups: its vanilla primitive-level configuration (used as the baseline in the main results) and an atom-level configuration that supplies the same components SIGS uses (used in the Section (v) ablation to isolate the latent-manifold contribution). SSDE uses its native primitive generator.

### E.1. Symbolic Baselines

*Table 22.* Baseline setups used in the paper.

| Method/protocol | Input supplied | Interpretation |
|---|---|---|
| HD-TLGP (atoms) | Atom-level knowledge base from the same symbolic families used by SIGS, followed by HD-TLGP's own discrete GP-style recombination, mutation, and coefficient tuning. | Compares search procedures while holding atom-level symbolic families fixed. |
| HD-TLGP (vanilla, primitive-level) | Primitive-level function set such as $+, -, \times, /, \sin, \cos, \exp, \tanh$. | Tests primitive-level symbolic discovery under the same PDE-residual scoring task. |
| SSDE | Tailored primitive dictionaries selected for each problem, using SSDE's reinforcement-learning symbolic generator. | Strong primitive-dictionary baseline with SSDE's native search procedure. |

#### E.1.1. HD-TLGP SETUP DETAILS

For the atom-level setup, the knowledge base contains atom-level pieces and partial components. The comparison holds atom-level symbolic families fixed while leaving assembly and coefficient optimization to HD-TLGP's discrete recombination machinery. The supplied knowledge-base entries are:

- **Diffusion**: first mode with exact amplitude $A\sin(\pi x/L)\exp(-\pi^2 Dt/L^2)$, templates for modes 3 and 5, and 2–3 mode combinations.

- **Burgers**: core shock $\tanh(\alpha(x - x_0 - st))$ and scaled variant.

- **Damped wave**: radial motif $\cos\left(k\sqrt{(x-x_0)^2 + (y-y_0)^2} - \omega t\right)$ and separable template $\sin(\pi x)\sin(\pi y)\cos(\omega t)\exp(-\gamma t)$.

- **Poisson–Gauss**: boundary mask $\sin(\pi x)\sin(\pi y)$, individual Gaussians for each center, and a sum-of-Gaussians template.

HD-TLGP then uses its own discrete GP-style recombination, mutation, and local coefficient tuning. This setup supplies symbolic ingredients and partial components while leaving completed assembly and coefficient selection to HD-TLGP; it is the same-symbolic-ingredient comparison with HD-TLGP's vanilla search procedure in place of SIGS's latent-manifold search.

The vanilla setup uses primitive-level expressions generated from elementary functions comparable to the SSDE primitive sets: $\{+, -, \times, \div, \sin, \cos, \exp, \tanh\}$ for 1D scalar problems, adding $\sqrt{\cdot}$ for radial damped waves and $\log$ for Poisson–Gauss. This tests primitive-level symbolic discovery under the same residual-scoring task. In both setups, we use HD-TLGP's standard population search with local constant optimization and the same wall-clock budgets reported in the main experiments.

The HD-TLGP settings are fixed across all runs except for population size, which is reduced for higher-dimensional spatial problems to fit the same wall-clock budget. We use population size 200 for 1D scalar problems and 50 for 2D problems,

*Table 26.* Discovered SIGS expressions for scalar benchmarks with known closed-forms.

| Problem | Discovered expression |
|---|---|
| Burgers | $0.86 + 0.6\tanh(30(x - 0.33 - 0.86t))$ |
| Diffusion | $3.974[\sin(2.15\pi x)e^{-3.21\pi^2 t} + \sin(0.71\pi x)e^{-0.36\pi^2 t} + \sin(3.58\pi x)e^{-8.93\pi^2 t}]$ |
| Damped Wave | $\cos(0.5\sqrt{(x+5)^2 + (y-5)^2} - 0.4t)e^{-0.45t}$ |

crossover probability 0.6, mutation probability 0.6, knowledge-base transfer probability 0.6 in the atom-level setup only, peephole simplification, and local constant optimization. Runs stop after 20 generations or the allotted wall-clock budget, whichever occurs first.

### E.2. SSDE Primitive Sets

*Table 23.* Primitive sets supplied to SSDE. These are elementary primitives used by SSDE's native symbolic generator; variables and numeric constants are included in all runs, and $x^n$ denotes low-degree polynomial powers up to $n = 3$.

| Problem | Primitive set |
|---|---|
| Burgers | $\{+, -, \times, /, \exp, \tanh, \sin, \cos\}$ |
| Diffusion | $\{+, -, \times, /, \exp, \tanh, \sin, \cos, \log\}$ |
| Damped Wave | $\{+, -, \times, /, \exp, \sin, \cos, \sqrt{\cdot}\}$ |
| PG-2/3/4 | $\{+, -, \times, /, \exp, \log, x^n, \sin, \cos\}$ |

SSDE uses its standard reinforcement-learning symbolic generator with learning rate $5 \times 10^{-4}$, entropy weight 0.07, batch size 1000, and 200,000 training samples. Expression length is constrained to 4–30 tokens, except for Diffusion where the upper bound is 60 to accommodate the multimode separated form. These settings are held fixed across the scalar benchmarks.

## F. Detailed Results and Discovered Expressions

This appendix groups detailed results according to the claims made in the main text. It first checks parity with prior-work benchmarks, then lists the expressions returned by SIGS and the baselines, and finally provides visual and numerical validation for scalar known-solution and Poisson–Gauss approximation cases.

### F.1. Parity Problems from Prior Work

*Table 24.* Problem specifications for the prior-work parity checks.

| Problem | Operator / residual | Dim. | Domain and mesh | Ground truth |
|---|---|---|---|---|
| Poisson1 (HD-TLGP) | $u_{xx} + u_{yy}$ | 2D | $[0, 1]^2$, $64^2$ | $\sin(\pi x)\sin(\pi y)$ |
| Advection3 (HD-TLGP) | $u_t + u_x + u_y$ | 2+1D | $[0, 1]^2 \times [0, 2]$, $64^2 \times 64$ | $\sin(x-t)+\sin(y-t)$ |
| Wave2D (SSDE) | $u_{tt} - (u_{xx} + u_{yy})$ | 2+1D | $[-1, 1]^2 \times [0, 1]$, $8^2 \times 8$ | $e^{x^2}\sin(y)e^{-0.5t}$ |

*Table 25.* Closed-form parity on prior-work problems. SIGS keeps symbolic constants such as $\pi$ as grammar terminals.

| Problem | Expression in prior work | SIGS expression |
|---|---|---|
| Poisson1 (HD-TLGP) | $\sin(3.141x)\sin(3.142y)$ | $\sin(\pi x)\sin(\pi y)$ |
| Advection3 (HD-TLGP) | $-\sin(0.9838t - x) - \sin(0.9979t - y)$ | $\sin(x - t) + \sin(y - t)$ |
| Wave2D (SSDE) | $e^{x^2 - 0.5t}\sin(y)$ | $e^{x^2}\sin(y)e^{-0.5t}$ |

### F.2. SIGS Discovered Expressions
### F.3. Scalar Solution Visualizations

The following figures provide the visual counterpart to the scalar known-solution results reported in the main text. Poisson–Gauss is defined and visualized separately in Section F.4, where the reference is a validated numerical solution.

*Table 27.* Discovered SIGS symbolic approximations for Poisson–Gauss. The search Ansatz allows a boundary-masked superposition; the table reports the signed exponential-quadratic terms retained in the fitted symbolic expression used for evaluation. Let $M(x, y) = \sin(\pi x)\sin(\pi y)$ and $B(A, \rho, c_x, c_y, d, s) = A\exp\big(\rho((x - c_x)^2 + (y - c_y)^2)/(d\,s^2)\big)$. Each row is $u_{\mathrm{SIGS}}(x, y) = M(x, y)\sum_j B_j(x, y)$.

| Problem | Retained $B(A, \rho, c_x, c_y, d, s)$ **terms** |
| --- | --- |
| PG-2 | $B(0.0080,\ 0.424,\ 0.923,\ 0.760,\ 2.136,\ 0.573)$ |
| | $+\, B(0.0251,\ -1.071,\ 0.794,\ 0.054,\ 2.245,\ 0.201)$ |
| | $+\, B(0.0105,\ -1.110,\ 0.248,\ 0.496,\ 1.862,\ 0.185)$ |
| PG-3 | $B(0.0079,\ 0.461,\ 0.500,\ -0.217,\ 2.152,\ 0.508)$ |
| | $+\, B(0.0137,\ -0.816,\ 0.750,\ 0.873,\ 1.898,\ 0.138)$ |
| | $+\, B(0.0137,\ -0.851,\ 0.250,\ 0.873,\ 2.505,\ 0.123)$ |
| | $+\, B(0.0206,\ -1.092,\ 0.500,\ 0.043,\ 1.738,\ 0.221)$ |
| PG-4 | $B(0.0068,\ -1.489,\ 0.731,\ 0.502,\ 1.553,\ 0.195)$ |
| | $+\, B(0.0112,\ -1.123,\ 0.500,\ 0.124,\ 1.804,\ 0.159)$ |
| | $+\, B(0.0294,\ -0.031,\ 0.665,\ 0.887,\ 2.025,\ 0.584)$ |
| | $+\, B(0.0069,\ -0.992,\ 0.267,\ 0.502,\ 1.664,\ 0.155)$ |
| | $+\, B(0.0286,\ -1.024,\ 0.501,\ -0.276,\ 1.569,\ 0.190)$ |

*Table 28.* Example baseline outputs illustrating the search behaviour behind the errors in the main tables. Long returned trees are shortened for readability.

| Problem | Method | Returned expression excerpt |
| --- | --- | --- |
| Burgers | SSDE | $\exp(\tanh(-1743.845x - 76821.176)/\sin(\exp(\tanh(\exp(x))))) \cdot \exp(\cdots)$ |
| Burgers | HD-TLGP (atom) | $\sin(\sin(\cos(\exp(-\tanh(0.996\tanh(25.8t - 30.0x + 9.9)))))) + 0.569$ |
| Diffusion | HD-TLGP (atom) | $3.974(\exp(88.121t + 3.974e^{-3.525t}\sin(63.495/x)\sin(2.249x))\sin(2.249x) + \sin(11.244x))e^{-91.646t}$ |
| PG-3 | SSDE | $x(-0.802y^2 + y)(1.092\cos(\sin(\sin(\sin(x)))) - 0.8246)$ |
| PG-4 | HD-TLGP (vanilla) | $9505.982$ |

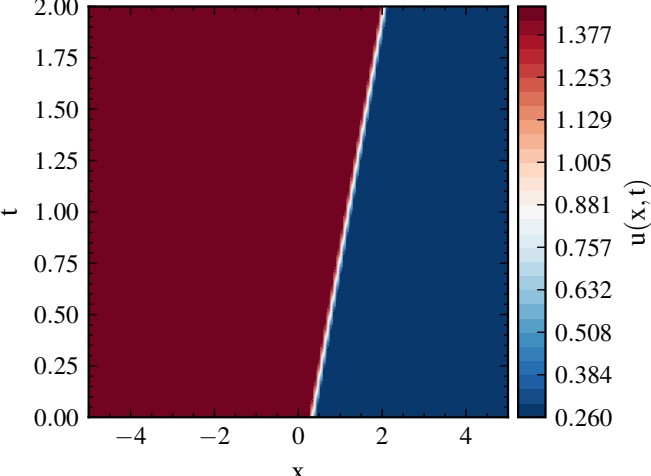

*Figure 5.* SIGS solution for the viscous Burgers equation on $x \in [-5, 5]$, $t \in [0, 2]$.

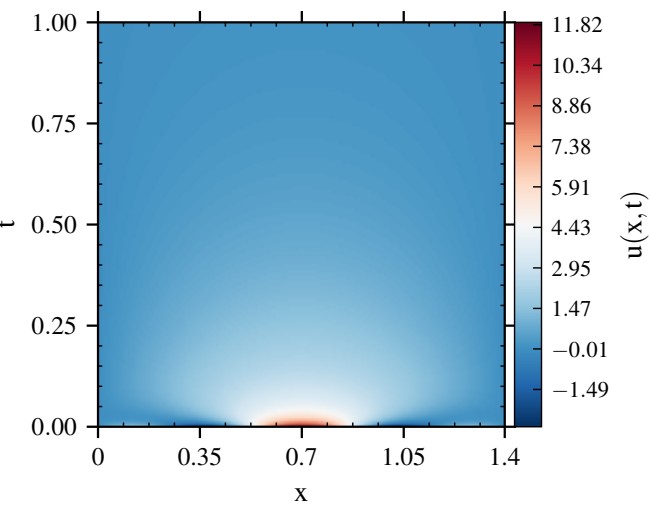

*Figure 6.* SIGS solution for the 1D diffusion equation on the evaluation grid.

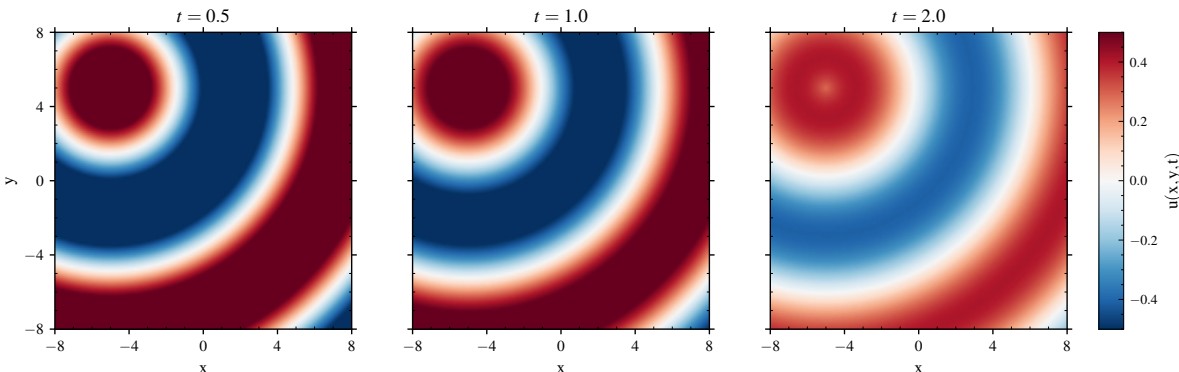

*Figure 7.* SIGS solution for the 2D damped-wave equation at selected time instances.

### F.4. Poisson–Gauss Problem Definition, Reference Validation, and Contours

For Poisson–Gauss, FEniCS provides validated numerical reference fields. This is the problem family used for the "no known closed-form" approximation results in the main text. On $\Omega = [0, 1]^2$, each PG-$n$ problem is

$$-\Delta u = f_n, \qquad u|_{\partial\Omega} = 0, \tag{15}$$

where $f_n$ is a sum of $n$ isotropic Gaussian sources with fixed width $\sigma = 0.1$:

$$f_n(x,y) = \sum_{i=1}^{n} \exp\left(-\frac{(x - \mu_{x,i})^2 + (y - \mu_{y,i})^2}{2\sigma^2}\right), \qquad \sigma = 0.1. \tag{16}$$

The source centers used in the plotted PG benchmarks are:

*Table 29.* Poisson–Gauss source configurations.

| Problem | Gaussian centers $(\mu_{x,i}, \mu_{y,i})$ |
|---|---|
| PG-2 | $(0.3, 0.5), (0.7, 0.2)$ |
| PG-3 | $(0.3, 0.8), (0.7, 0.8), (0.5, 0.2)$ |
| PG-4 | $(0.3, 0.5), (0.7, 0.5), (0.5, 0.2), (0.5, 0.7)$ |

The selected high-level Ansatz is the generic masked superposition

$$u(x, y) = \sin(\pi x)\sin(\pi y)\sum_{j=1}^{K} a_j\phi_j(x, y), \tag{17}$$

with $K = 3, 4, 8$ for PG-2, PG-3, and PG-4, respectively. These $K$ values are fixed before search as approximation budgets that increase with the number of source centers. The sine mask enforces the homogeneous Dirichlet boundary condition, while the $\phi_j$ are selected from the same fixed atom corpus as all other experiments. We validate the FEniCS reference by mesh refinement and by checking the weak-form energy identity. Symbolic expressions are compared to the FEniCS solution by projecting the expression into the finite-element space and computing relative $L^2$ error.

The finite-element reference uses continuous $P2$ elements on refined triangular meshes. The weak form is: find $u_h \in V_h$ with $u_h|_{\partial\Omega} = 0$ such that

$$a(u_h, v_h) = L(v_h) \quad \forall v_h \in V_h, \qquad a(u, v) = \int_\Omega \nabla u \cdot \nabla v\, dx, \quad L(v) = \int_\Omega fv\, dx. \tag{18}$$

As a consistency check, the computed solution satisfies the energy identity $a(u_h, u_h) = L(u_h)$ up to solver tolerance, and the strong-form residual decreases under mesh refinement. For a symbolic candidate $u_{\mathrm{sym}}$, we compute its $L^2$ projection $\Pi_h u_{\mathrm{sym}} \in V_h$ and report

$$\frac{\|u_h - \Pi_h u_{\mathrm{sym}}\|_{L^2(\Omega)}}{\|u_h\|_{L^2(\Omega)}}. \tag{19}$$

Figures 8–10 all use the same panel order: left, the source field $f_n$; middle, the FEniCS reference $u_h$; right, the SIGS symbolic approximation evaluated on the same visualization grid. The source panel has its own colour scale, while the FEniCS and SIGS solution panels are plotted on matched solution scales so that boundary behaviour and smoothed interior structure can be compared directly.

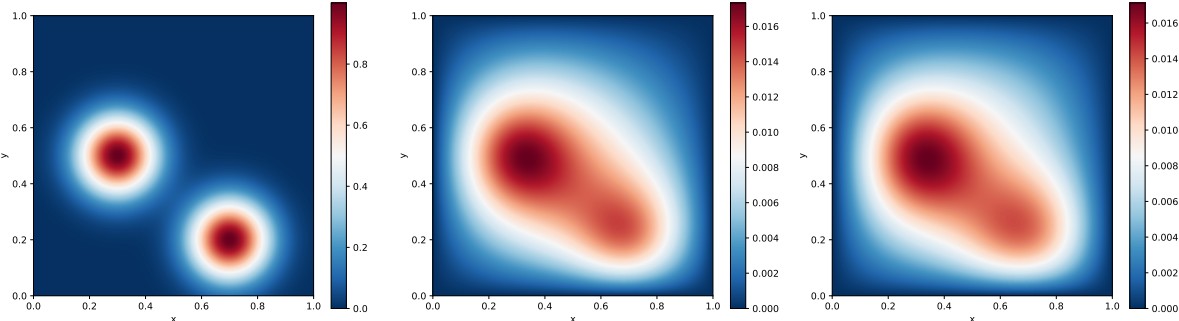

*Figure 8.* Poisson–Gauss PG-2 on $[0, 1]^2$. Left: $f_2$, the sum of Gaussian sources centered at $(0.3, 0.5)$ and $(0.7, 0.2)$ with $\sigma = 0.1$. Middle: FEniCS $P2$ finite-element reference $u_h$. Right: SIGS symbolic approximation $u_{\mathrm{SIGS}}$, whose relative $L^2$ error against the projected FEniCS reference is 2.66%.

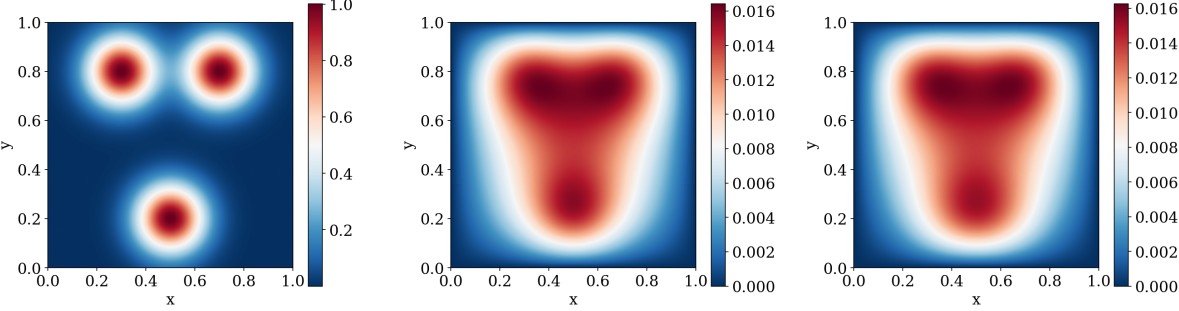

*Figure 9.* Poisson–Gauss PG-3. Left: $f_3$, with Gaussian centers $(0.3, 0.8)$, $(0.7, 0.8)$, and $(0.5, 0.2)$, all with $\sigma = 0.1$. Middle: FEniCS $P2$ reference. Right: SIGS symbolic approximation. The matched middle/right colour scales show that the boundary-masked symbolic expression captures the Green-smoothed three-source structure; the relative $L^2$ error is 1.54%.

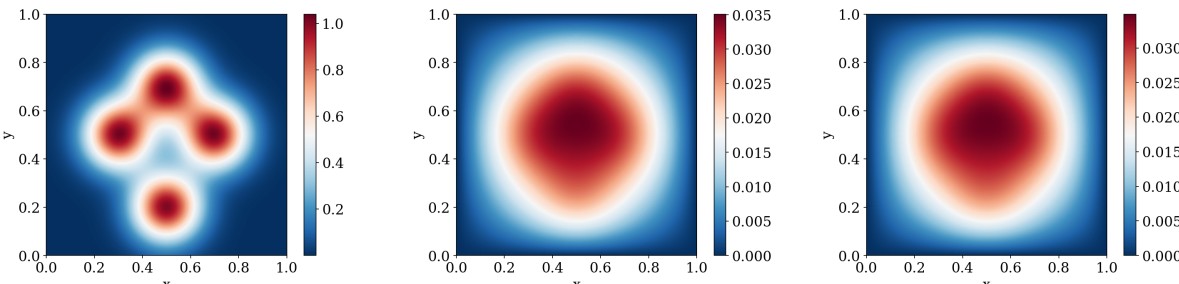

*Figure 10.* Poisson–Gauss PG-4. Left: $f_4$, with Gaussian centers $(0.3, 0.5)$, $(0.7, 0.5)$, $(0.5, 0.2)$, and $(0.5, 0.7)$, all with $\sigma = 0.1$. Middle: FEniCS $P2$ reference. Right: SIGS symbolic approximation. The relative $L^2$ error is $1.05\%$, illustrating structured approximation against a validated numerical reference.

*Table 30.* FEniCS residual convergence for PG-2. The monotone decrease supports using the numerical solution as the reference for relative-error evaluation.

| Mesh | DOF | Runtime (s) | PDE residual |
|------|-----|-------------|--------------|
| $32 \times 32$ | 4,225 | 0.067 | $2.0525 \times 10^{-2}$ |
| $64 \times 64$ | 16,641 | 0.210 | $1.0238 \times 10^{-2}$ |
| $128 \times 128$ | 66,049 | 0.946 | $5.1154 \times 10^{-3}$ |
| $256 \times 256$ | 263,169 | 3.550 | $2.5573 \times 10^{-3}$ |
| $512 \times 512$ | 1,050,625 | 14.920 | $1.2787 \times 10^{-3}$ |

*Table 31.* Residual comparison for PG symbolic approximations and FEniCS references on two meshes. SIGS is a fixed expression, so its residual remains approximately constant under mesh refinement.

| Mesh | Residual | PG-2 | PG-3 | PG-4 |
|------|----------|------|------|------|
| $128^2$ | $R(u_{\text{FEniCS}})$ | $2.924 \times 10^{-4}$ | $3.663 \times 10^{-4}$ | $4.070 \times 10^{-4}$ |
| $128^2$ | $R(u_{\text{SIGS}})$ | $4.491 \times 10^{-2}$ | $4.476 \times 10^{-2}$ | $3.617 \times 10^{-2}$ |
| $256^2$ | $R(u_{\text{FEniCS}})$ | $8.048 \times 10^{-5}$ | $1.034 \times 10^{-4}$ | $1.133 \times 10^{-4}$ |
| $256^2$ | $R(u_{\text{SIGS}})$ | $4.493 \times 10^{-2}$ | $4.490 \times 10^{-2}$ | $3.618 \times 10^{-2}$ |

## G. Ablation and Robustness Details

The main text groups all five ablations into a single section. Here we state what is fixed, what is changed, and what each result supports. Table 32 follows the SIGS machinery in pipeline order: primitive CFG expansion, atom-level candidate construction, latent-manifold search, and TGVAE topology. Table 33 records the admissible-prior and grammar-completeness checks: redundant Ansatz slots, controlled primitive removal with an equivalent representation, and broad symbolic approximation.

*Table 32.* Mechanism comparisons in pipeline order. The first two rows compare candidate-space construction and search procedure; the third measures geometry regularisation.

| Mechanism | Fixed | Changed | Finding |
|-----------|-------|---------|---------|
| Primitive-level CFG | Damped-wave PDE, residual, and grid | primitive-level CFG proposals | 50,000 samples give 133 admissible candidates; best relative error is about $366\%$. |
| Latent manifold vs. discrete tree search | atom-level symbolic in-gredients and residual objective | SIGS latent search vs HD-TLGP tree search over the same atoms | The atom-level tree search remains at $2.04\%$, $33.34\%$, and $423.30\%$ on Burgers/Diffusion/Damped Wave, while SIGS reaches $10^{-13}$-scale error. |
| TGVAE topology | corpus, decoder, and off-training sampling task | GVAE vs TGVAE geome-try regularisation | TGVAE needs $1433.2 \pm 27.3$ attempts for 1000 valid decodes vs $1486.2 \pm 19.5$ for GVAE, a $3.56\% \pm 1.81$ reduction. |

*Table 33.* Admissible-prior sensitivity checks. These rows separate overspecified Ansätze, equivalent-form primitive removal, and approximation without a closed-form target.

| Check | Fixed | Changed | Finding |
|---|---|---|---|
| Ansatz over-specification | Burgers PDE, grammar, and one-front reference | enlarge the sum Ansatz from the minimal $K = 1$ to $K = 2, 3, 5$ slots | Redundant coefficients are suppressed; machine precision for $K \leq 3$, $4.52 \times 10^{-5}$ at $K = 5$. |
| Equivalent primitive removal | KdV PDE and polynomial-in-atom Ansatz | remove cosh and sech, while tanh remains available | SIGS finds the equivalent $2 - 2\tanh^2(x - 4t)$ form with $6.6 \times 10^{-6}$ error. |
| Broad approximation prior | Poisson–Gauss PDE, Dirichlet boundary mask, and fixed corpus | no closed-form target; evaluate against FEniCS | 1–3% symbolic approximation using the boundary-masked superposition Ansatz. |

## G.1. Burgers Ansatz Over-Specification

The Burgers solution requires one transition-layer atom. To test sensitivity to redundant slots, we use an enlarged sum Ansatz with $K$ atom terms and allow coefficient refinement to suppress unnecessary terms.

*Table 34.* Burgers over-specification experiment. The true solution requires $K = 1$.

| Number of slots $K$ | Interpretation | Relative $L^2$ error |
|---|---|---|
| 1 | matched minimal Ansatz | $\approx 10^{-13}$ |
| 2 | one redundant slot | $\approx 10^{-13}$ |
| 3 | two redundant slots | $\approx 10^{-13}$ |
| 5 | four redundant slots | $4.52 \times 10^{-5}$ |

## G.2. KdV Primitive Removal

The Korteweg–de Vries equation is the nonlinear dispersive PDE

$$u_t + 6uu_x + u_{xxx} = 0, \tag{20}$$

posed here on $[-10, 10] \times [0, 1]$. The one-soliton reference solution is

$$u^\star(x, t) = \frac{2}{\cosh^2(x - 4t)}. \tag{21}$$

In the misspecification experiment, cosh and sech are removed from the grammar, while tanh remains available. Under the polynomial-in-one-atom Ansatz

$$u(x, t) = \sum_{k=0}^{2} a_k \phi(x, t)^k, \tag{22}$$

SIGS returns

$$u_{\text{SIGS}}(x, t) = 0.0003734 \tanh(0.001355t - 0.0006419x + 0.0002936)$$
$$- 2.0 \tanh^2(4.0t - x + 2.237 \times 10^{-7}) + 2.0, \tag{23}$$

whose dominant term is

$$u_{\text{SIGS}}(x, t) \approx 2 - 2\tanh^2(x - 4t), \tag{24}$$

with relative $L^2$ error $6.6 \times 10^{-6}$. This is an equivalent in-grammar representation by the hyperbolic identity

$$2\operatorname{sech}^2(z) = 2(1 - \tanh^2(z)) = 2 - 2\tanh^2(z). \tag{25}$$

The experiment demonstrates equivalent-form recovery under controlled primitive removal.

## H. Coupled Nonlinear Systems

This appendix gives the full coupled-system specifications behind the main-text nonlinear results. Shallow Water is used as a high-accuracy coupled recovery benchmark, while Compressible Euler is treated as a harder compact symbolic-approximation benchmark over four interacting fields.

## H.1. Shallow Water Equations

We use a manufactured 2D shallow-water system with density $\rho$ and velocities $(u, v)$ on $[-10, 10]^2 \times [0, 5]$. In the same residual notation as Appendix D.1, $U = (\rho, u, v)$ and $\mathcal{S}_{\text{SWE}}[U] - f = 0$:

$$\rho^\star(x, y, t) = 1 + h \exp\left(-\frac{r}{w(1+t)}\right)\cos(k\sqrt{r} - ct)e^{-\alpha t}, \qquad r = (x - x_0)^2 + (y - y_0)^2. \tag{26}$$

The manufactured velocities are tied to the density through shallow-water radial transport factors,

$$u^\star(x, y, t) = \rho^\star(x, y, t)\frac{x\,c}{H\sqrt{r + \varepsilon}}, \qquad v^\star(x, y, t) = \rho^\star(x, y, t)\frac{y\,c}{H\sqrt{r + \varepsilon}}, \tag{27}$$

with $\varepsilon > 0$ used only to avoid division by zero at the center. The velocity factors are evaluated directly from these expressions, with the center $(x_0, y_0)$ entering through $r$. The forcing terms below are obtained by substituting these manufactured fields into the PDE.

$$\rho_t + (\rho u)_x + (\rho v)_y - f_\rho(x, y, t) = 0,$$
$$(\rho u)_t + \left(\rho u^2 + \tfrac{1}{2}g\rho^2\right)_x + (\rho u v)_y - f_u(x, y, t) = 0, \tag{28}$$
$$(\rho v)_t + (\rho u v)_x + \left(\rho v^2 + \tfrac{1}{2}g\rho^2\right)_y - f_v(x, y, t) = 0.$$

The run uses $h = 0.97$, $w = 0.88$, $k = 1.78$, $c = 1.28$, $\alpha = 0.72$, $(x_0, y_0) = (3.77, 2.34)$, and $H = 4.46$, with periodic boundary conditions and initial conditions taken from the manufactured fields at $t = 0$. The Ansatz couples the fields through density: $\rho = \phi_1 \phi_2 \phi_3$, $u = \rho\psi_x$, and $v = \rho\psi_y$. SIGS obtains relative $L^2$ errors $1.8731 \times 10^{-4}$ for $\rho$, $2.2310 \times 10^{-4}$ for $u$, and $4.1783 \times 10^{-4}$ for $v$, with runtime 3m23s.

*Table 35.* Manufactured shallow-water constants used in the system above.

|       | $h$  | $w$  | $k$  | $c$  | $\alpha$ | $x_0$ | $y_0$ | $H$  | $g$  |
|-------|------|------|------|------|------|------|------|------|------|
| Value | 0.97 | 0.88 | 1.78 | 1.28 | 0.72 | 3.77 | 2.34 | 4.46 | 9.81 |

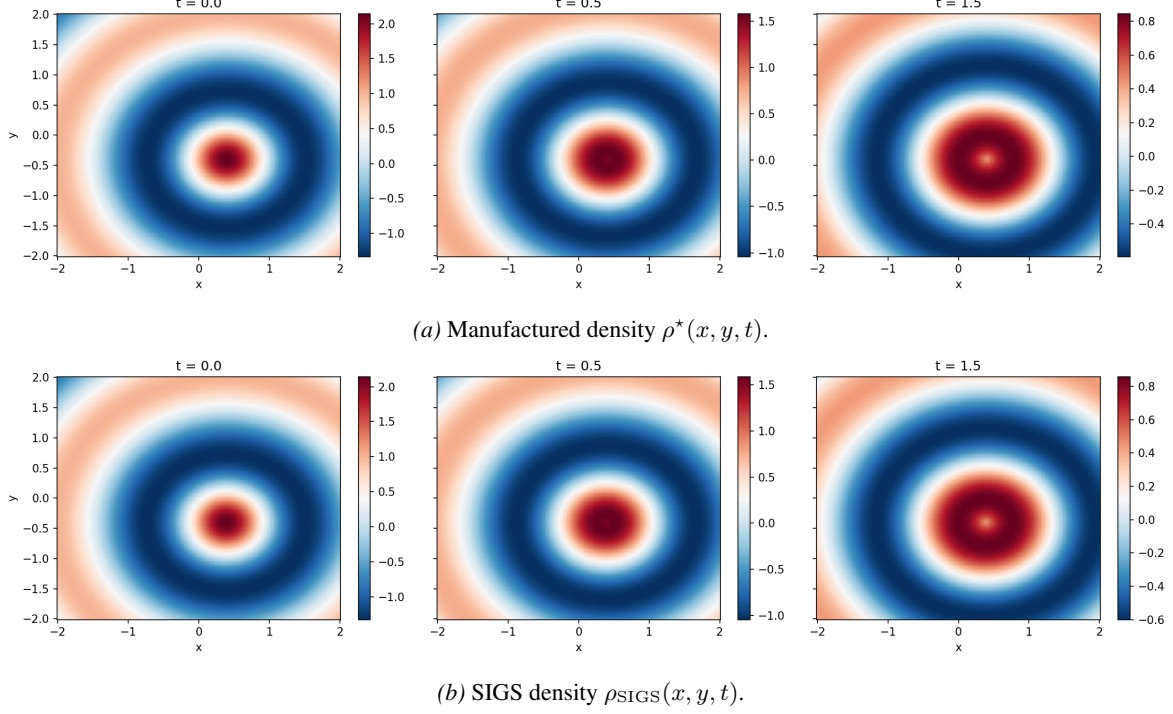

*(a)* Manufactured density $\rho^\star(x, y, t)$.

*(b)* SIGS density $\rho_{\text{SIGS}}(x, y, t)$.

*Figure 11.* Shallow-water density fields at representative times.

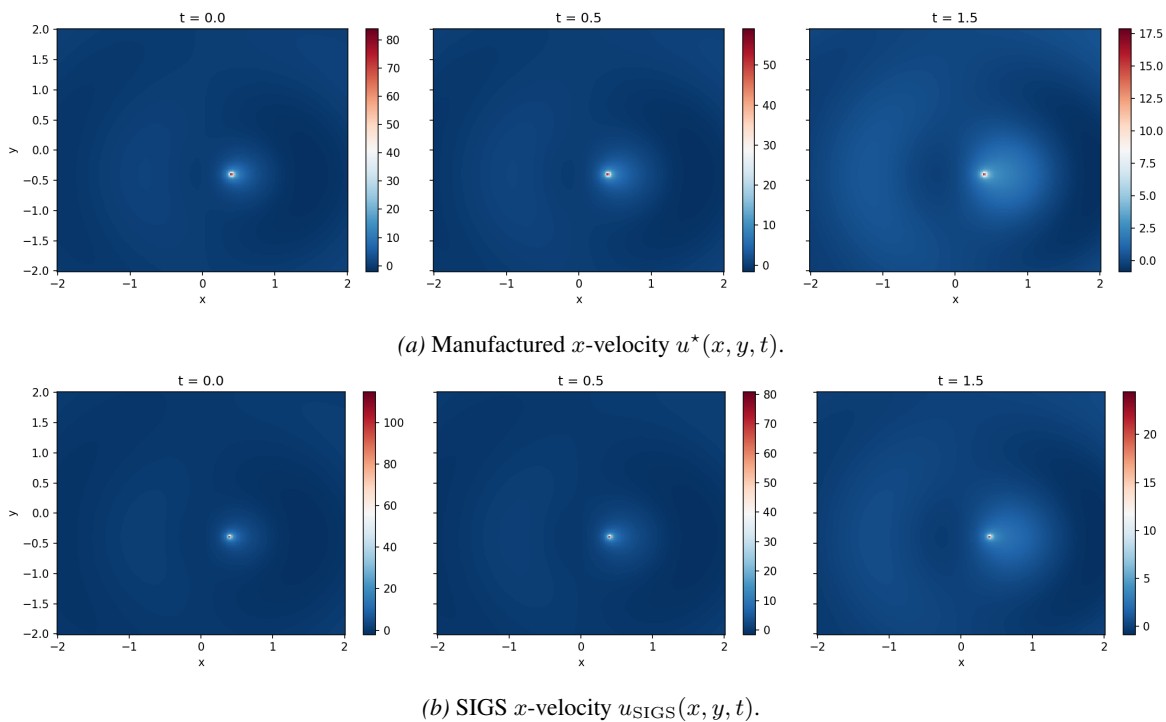

*(a)* Manufactured $x$-velocity $u^\star(x, y, t)$.

*(b)* SIGS $x$-velocity $u_{\text{SIGS}}(x, y, t)$.

*Figure 12.* Shallow-water $x$-velocity fields at representative times.

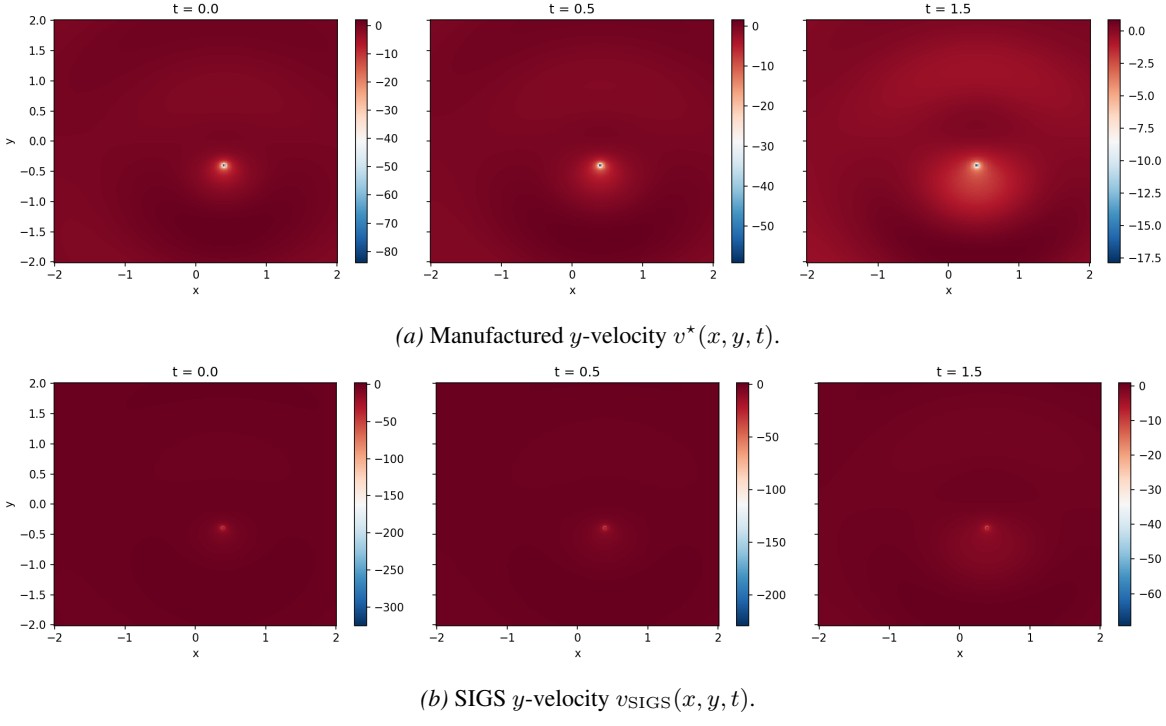

*(a)* Manufactured $y$-velocity $v^\star(x, y, t)$.

*(b)* SIGS $y$-velocity $v_{\text{SIGS}}(x, y, t)$.

*Figure 13.* Shallow-water $y$-velocity fields at representative times.

## H.2. Compressible Euler

For Compressible Euler, SIGS solves a harder coupled symbolic approximation task. We use a steady manufactured system on the periodic domain $[0,1]^2$. The primitive unknowns are $U = (\rho, u, v, p)$; the conservative state is $(\rho, \rho u, \rho v, E)$, with total energy $E = p/(\gamma - 1) + \frac{1}{2}\rho(u^2 + v^2)$ and $\gamma = 1.4$. In residual form,

$$\mathcal{S}_{\mathrm{CE}}[U] - f = \partial_x F_x(U) + \partial_y F_y(U) - \begin{pmatrix} f_\rho & f_u & f_v & f_E \end{pmatrix}^\top = 0, \tag{29}$$

where

$$F_x(U) = \begin{pmatrix} \rho u \\ \rho u^2 + p \\ \rho u v \\ (E + p)u \end{pmatrix}, \qquad F_y(U) = \begin{pmatrix} \rho v \\ \rho u v \\ \rho v^2 + p \\ (E + p)v \end{pmatrix}.$$

The forcing $(f_\rho, f_u, f_v, f_E)$ is obtained by substituting the manufactured fields of Table 36 into the residual (29); explicit closed-form expressions are in the released code. Equivalently, the four residual equations enforce mass, $x$-momentum, $y$-momentum, and energy balance with manufactured forcing. Periodic boundary conditions are imposed on $\rho, u, v, p$. The Ansatz uses six spatial atom slots per field, with exponential envelopes for $\rho$ and $p$:

$$\rho = e^{\sum_{i=1}^{6} f_i(x,y)}, \qquad u = \sum_{i=1}^{6} g_i(x, y), \qquad v = \sum_{i=1}^{6} h_i(x, y), \qquad p = e^{\sum_{i=1}^{6} k_i(x,y)}. \tag{30}$$

The per-field relative errors are 10.8% for $\rho$, 9.93% for $u$, 9.84% for $v$, and 12.1% for $p$. We report CE as a compact symbolic approximation.

The coefficient-level manufactured and optimized fields are shown in Table 36, where $s_{ij} = \sin(i\pi x)\sin(j\pi y)$ denotes the product sine modes used to express the manufactured and optimized fields. The released code includes the CE residual, forcing construction, and evaluation scripts.

*Table 36.* Compressible Euler manufactured and SIGS-optimized field forms. Coefficients are truncated to three significant figures.

| Field | Manufactured | SIGS optimized | Rel. $L^2$ |
|---|---|---|---|
| $\rho$ | $\exp[-0.0887s_{11} + 0.504s_{12} + 0.259s_{21} + 0.140s_{22}]$ | $\exp[0.273s_{12} + 0.217s_{21} + 0.229s_{22} + 2.18\times10^{-3}\sin(2\pi y)\cos(\pi x)]$ | 10.8% |
| $u$ | $\pi[-0.243s_{11} - 0.385s_{12} - 0.494s_{21} + 0.518s_{22}]$ | $-0.929s_{11} - 1.00s_{12} + 0.0518\sin(\pi x)\cos(3\pi y) - 1.50s_{21} + 1.56s_{22}$ | 9.93% |
| $v$ | $\pi[0.0715s_{11} + 0.233s_{12} - 0.536s_{21} + 0.665s_{22}]$ | $0.718s_{12} + 2.05s_{22} - 0.307\sin(4\pi x)\sin(\pi y) - 1.22\sin(\pi y)\cos(\pi x) + 1.00\sin(\pi y)\cos(3\pi x)$ | 9.84% |
| $p$ | $\exp[0.235s_{11} - 0.322s_{12} - 0.356s_{21} - 0.448s_{22}]$ | $\exp[-0.389s_{12} - 0.354s_{21} - 0.410s_{22}]$ | 12.1% |

*Table 37.* SIGS performance on coupled nonlinear PDE systems. SWE is evaluated as high-accuracy coupled recovery; CE is evaluated as coupled symbolic approximation.

| System | Field | Relative $L^2$ error | Runtime |
|---|---|---|---|
| Shallow Water | $\rho$ | $1.87 \times 10^{-4}$ | |
| | $u$ | $2.23 \times 10^{-4}$ | 3m23s |
| | $v$ | $4.18 \times 10^{-4}$ | |
| Compressible Euler | $\rho$ | 10.8% | |
| | $u$ | 9.93% | 5m12s |
| | $v$ | 9.84% | |
| | $p$ | 12.1% | |

## I. Classical Tools, LLMs, and Runtime

Classical tools and LLM outputs provide additional context on the same problem specifications. They show the difference between returning a formally correct representation, returning a compact usable expression, and returning an expression that satisfies the PDE and conditions.

### I.1. Computer Algebra

We used Mathematica `DSolveValue` on the same problem specifications. Table 38 summarizes whether the returned representation is compact and directly usable for the comparison tasks.

*Table 38.* Classical-tool calibration on selected problems.

| Problem | Mathematica | ChatGPT-style reasoning |
|---|---|---|
| Burgers | No compact solution | Recovers manufactured shock form |
| Damped wave | No compact solution | PDE specification check |
| Diffusion | Infinite Fourier series | PDE specification check |
| Poisson–Gauss (PG-2) | Green's-function / infinite-series form | Boundary-violating approximation |
| KdV primitive-removal case | No compact expression returned under the tested setup | PDE specification check |
| Coupled SWE/CE | Outside tested compact `DSolveValue` mode | PDE specification check |

For the Dirichlet heat equation, Mathematica returns the standard sine-series representation

$$u(x,t) = \sum_{n=1}^{\infty} b_n \exp\left[-D\left(\frac{n\pi}{L}\right)^2 t\right] \sin\left(\frac{n\pi x}{L}\right), \qquad b_n = \frac{2}{L}\int_0^L u_0(\xi)\sin\left(\frac{n\pi\xi}{L}\right) d\xi. \tag{31}$$

This is the classical infinite-series solution form. For PG-2, Mathematica returns a Green's-function representation

$$u(x,y) = 2\sum_{n=1}^{\infty} \sin(n\pi x)\int_0^1 G_n(y,\eta)\left(\int_0^1 \sin(n\pi\xi)f(\xi,\eta)d\xi\right) d\eta, \tag{32}$$

where $G_n$ is the one-dimensional Dirichlet Green's function for $\partial_{yy} - (n\pi)^2$. This representation is an infinite modal integral expression.

### I.2. LLM Calibration

We also tested an extended-reasoning LLM (GPT-5.1 with extended thinking) system by providing the full PDE specification, domain, and boundary/initial conditions. On Burgers, the model recovered the manufactured traveling-shock form

$$u_{\text{LLM}}(x,t) = 0.86 - 0.6\tanh(30x - 25.8t - 9.9), \tag{33}$$

which is consistent with recognizing the `tanh`-shaped manufactured data. On PG-2, it returned a plausible free-space Gaussian-potential expression,

$$\begin{aligned}
u_{\text{LLM}}(x,y) = -0.01\Big( &\log\sqrt{(x-0.3)^2 + (y-0.5)^2} - \tfrac{1}{2}\,\text{Ei}\Big(-\frac{(x-0.3)^2 + (y-0.5)^2}{0.02}\Big) \\
+ &\log\sqrt{(x-0.7)^2 + (y-0.2)^2} - \tfrac{1}{2}\,\text{Ei}\Big(-\frac{(x-0.7)^2 + (y-0.2)^2}{0.02}\Big)\Big),
\end{aligned} \tag{34}$$

whose boundary trace is nonzero under the homogeneous Dirichlet condition. Its relative $L^2$ error against the FEniCS reference is 157.6%.

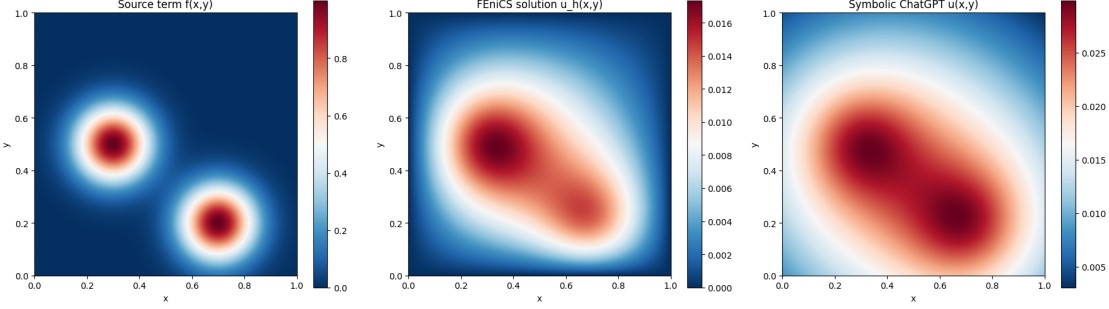

*Figure 14.* ChatGPT-style output on PG-2: source term, FEniCS reference, and boundary-violating proposed expression.

*Table 39.* ChatGPT-style calibration metrics.

| Problem | Relative $L^2$ error | Reasoning time |
|---|---|---|
| Burgers | 0.0% | 7m25s |
| Poisson–Gauss (PG-2) | 157.6% | 8m16s |

*Table 40.* SIGS runtime on representable scalar PDEs run to float64 machine-precision tolerance, a stricter target than Table 5 that no baseline reaches within budget.

| Problem | Relative $L^2$ error | Total SIGS time |
|---|---|---|
| Burgers | $6.64 \times 10^{-14}$ | 13.35s |
| Diffusion | $7.16 \times 10^{-13}$ | 39.2s |
| Damped Wave | $1.22 \times 10^{-13}$ | 30.2s |

**LLM usage.** The authors used LLM-based tools for grammar and spelling polish. The scientific claims, experiments, and camera-ready changes were checked by the authors.

**Code availability.** Code and reproduction scripts are available at `https://github.com/oroikono/SIGS`.

