# OpenReview forum: "Neuro-Symbolic AI for Analytical Solutions of Differential Equations"
_ICML.cc/2026/Conference — ICML 2026 regular_

### Official Review · Reviewer_Efpo · 2026-02-21

**Soundness:** 2
**Presentation:** 3
**Significance:** 2
**Originality:** 2
**Overall Recommendation:** 2
**Confidence:** 4

**Summary:**

The paper introduces SIGS (Symbolic Iterative Grammar Solver), a neuro-symbolic framework for discovering closed-form analytical solutions to PDEs. The method uses a context-free grammar to generate mathematical building blocks (atoms), embeds them into a continuous latent space via a Grammar Variational Autoencoder (GVAE), and searches this space using iterative clustering followed by gradient-based coefficient refinement. The authors claim state-of-the-art performance on several PDE benchmarks, including coupled nonlinear systems and cases without known closed-form solutions.

**tl;dr** If the partial differential equation has a closed-form solution that *could* be constructed using pre-defined grammar, we could significantly outperform numerical methods. But how many PDEs have such one?

**Compliance With Llm Reviewing Policy:**

Affirmed.

**Final Justification:**

In paper proposed method that uses PINN-like approach but using "symbolic regression", so the output is the solution of initial-boundary value problem for PDE.

Whereas PINN and SIGS definetely are different, they are mirror and should be compared. The main reason of score is the contradiction betweenf universal PINN have 2-5%, sometimes 10-20% error and non-universal symbolic expressions meet numerical precision.

There are no miracles, we pay for precision with non-generalizability, and more general with bias. Also we pay with that fact that the expert is (as everywhere and it is noted and agreed) required. The difference is that in SIGS we require rare mathematical physics expert (and I insist that for some cases, for example wave equation in polar coordinates, where Bessel functions are required. It should be not an ordinary mathematics expert, that should make sure that at least $\frac{1}{\sqrt{z}}\sin(z)$ should be reachable within grammatics) to define grammatics and in PINN case we require rather machine learning expert such are more common these days.

So we insist that there are no barriers to compute obtained expressions on the grid, to get the numbers and compare with PINNs using PINNacle - this would be the most convincing point and could not be done within short rebuttal period. However, we note and mention that the paper has its interest, even though is too niche. With a significant development it could serve to understand how PINNs work. As authors mentioned "one cannot tell if it is architecture limitation or optimization failure" and much more, if it is collocation point problems, loss reweighting problem - those four are definitely open quesion in a domain.

**Key Questions For Authors:**

1. The grammar for each benchmark contains atoms that directly match the known solution structure. How would SIGS perform if the correct atom class were entirely absent from the grammar, rather than the single primitive omitted in the KdV experiment? A systematic evaluation of grammar misspecification would substantially strengthen the robustness claims.
2. The ablation (experiment vi) shows near-total failure without pre-specified atoms. This suggests the latent search provides minimal value beyond coefficient fitting. What evidence supports that the GVAE search is doing meaningful work beyond identifying the correct atom family?
3. No comparison with modern symbolic regression methods (PySR, LLM-SR, deep symbolic regression) is provided. These methods can be equipped with PDE residual fitness functions without requiring hand-crafted atoms. Why were they excluded, and how does SIGS compare?
4. The entire evaluation uses noiseless, manufactured data. MDBench and PDEBench provide real trajectory data for several overlapping problems under noise. How do SIGS's discovered expressions perform when evaluated against such data?
5. The Ansatz design is described as user-specified but the effort required is not documented. For a new PDE outside the benchmark suite, how much expert knowledge is required to design an appropriate grammar and Ansatz, and how sensitive are results to these choices?

**Limitations:**

Partially addressed. The paper acknowledges scalability limitations and grammar dependency, but does not adequately discuss the Ansatz design burden, the restriction to noiseless settings, or the narrow class of PDEs to which the method applies in practice.

**Strengths And Weaknesses:**

The paper presents a working engineering system with genuinely clean implementation, and the coupled PDE results (Shallow Water, Compressible Euler) are the most credible contribution. However, the central evaluation is circular, key baselines from the symbolic regression literature are missing, robustness to noise is never tested, and the comparison against Mathematica is qualitative only. In its current form the empirical claims are not adequately supported. A resubmission to a domain-specific venue with honest framing of the method's scope, proper baselines, and noise robustness evaluation would be appropriate.

**Overall Strengths**

- The problem is well-motivated and practically relevant. Analytical solutions to PDEs provide genuine scientific value that numerical methods cannot replicate.
- The pipeline is cleanly engineered and the paper is well written. The combination of grammar-constrained generation, latent space search, and coefficient refinement is coherent and the implementation appears careful.
- The coupled PDE experiments (Shallow Water Equations, Compressible Euler) are the most credible contribution. Baselines genuinely fail on these problems and the results, while not perfect, demonstrate that grammar-guided search can handle multi-field dependencies that previous symbolic methods cannot express.
- The topological regularization of the latent space is a technically interesting addition that could have independent value beyond this specific application.
- The KdV grammar misspecification experiment is an honest test and the result (recovering an equivalent form via trigonometric identities) is genuinely interesting.

**Overall Weaknesses**

- The evaluation is largely circular. The grammar is designed to contain atoms matching the benchmark solution structures, making the main results (Tables 3 and 5) closer to structured coefficient fitting than autonomous solution discovery.
- The ablation inadvertently confirms this: without pre-specified atoms the method collapses entirely, revealing that the latent search machinery contributes little independently of the grammar design.
- Key baselines from the modern symbolic regression literature (PySR, LLM-SR, deep symbolic regression with physics-informed fitness) are absent.
- The method is never tested under noise or on real trajectory data, which is a significant gap given the practical framing.
- The comparison against Mathematica is qualitative only, which is insufficient for a paper claiming to automate analytical solution discovery.
- The Ansatz and grammar design effort per problem is substantially understated, limiting reproducibility and practical applicability.

## Soundness

The central soundness concern is that the evaluation is largely circular. The grammar is explicitly constructed to contain atoms matching the solution structures of the benchmark problems. For Burgers, the grammar directly encodes shock transition profiles of the form $\tanh((x - x_0 - st)(u_l - u_r)/4\nu)$, which is structurally identical to the discovered solution. Similarly, the diffusion and wave equation atoms are eigenfunction families specifically designed for those operators. Given this, the impressive results in Tables 3 and 5 primarily demonstrate that gradient-based coefficient fitting works well when the solution structure is pre-specified, rather than that SIGS autonomously discovers solutions.

This concern is inadvertently confirmed by the ablation study (Section 3, experiment vi). When atoms are replaced by primitive rules, performance collapses completely: only 133 out of 50,000 sampled functions were admissible and the best achieved roughly 366% relative error. The authors present this as evidence that atoms are important, which is technically true, but it simultaneously reveals that the latent search and GVAE machinery contribute almost nothing without correctly pre-specified atoms. The method is essentially a sophisticated coefficient fitter wrapped around expert-designed solution templates.

A further soundness issue is the absence of robustness testing. The entire framework assumes perfect knowledge of the PDE operator and noiseless conditions. Real-world applications invariably involve measurement noise and model uncertainty. Recent benchmarks such as MDBench (Bideh et al., AAAI 2026) provide trajectory data for several overlapping problems (Burgers, KdV, Advection) under varying noise levels. Evaluating SIGS's discovered expressions against such data would provide a much more honest picture of practical utility, and is a natural experiment the paper avoids entirely.

## Presentation

The paper is clearly written and the figures are informative. The overview figure (Figure 1) effectively communicates the pipeline. However, the Ansatz design effort per problem is substantially understated. The paper does not provide sufficient detail for a practitioner to reproduce the grammar and Ansatz design for a new PDE, which is arguably the hardest and most expert-intensive part of the method. Reproducibility is therefore limited.

## Significance

The problem of finding closed-form solutions to PDEs is genuinely valuable. Analytical solutions provide interpretability, exact parameter dependencies, and physical insight that numerical methods cannot. However, the class of PDEs admitting closed-form solutions expressible by a grammar of elementary functions is inherently narrow. The benchmarks are predominantly manufactured problems designed around known solution structures, which limits the demonstration of practical significance.

The comparison against classical baselines is also insufficient. Mathematica is dismissed with a qualitative pass/fail table (Table 24) without runtime or accuracy numbers on the same problems. For a paper claiming to automate analytical solution discovery, a rigorous quantitative comparison against the leading Computer Algebra System is essential.

## Originality

The GVAE for symbolic expressions is directly adopted from Kusner et al. (2017). The topological regularization (hull loss, persistent homology, smoothness penalty) is a technically interesting incremental addition but is not evaluated independently as a contribution. The iterative subclustering search is reasonable but not conceptually novel. The grammar-based approach to generating valid mathematical expressions addresses a real problem but is a natural engineering solution rather than a deep methodological advance.

Most importantly, the paper omits comparison with the modern symbolic regression literature. Methods such as LLM-SR, PySR with physics-informed fitness functions, and deep symbolic regression approaches are natural baselines that the paper does not consider. These methods can incorporate physical priors without requiring hand-crafted atom libraries, which directly challenges the paper's central motivation. The choice of baselines (HD-TLGP, SSDE) appears to favor methods particularly vulnerable to the combinatorial explosion argument, rather than representing the current state of the art.

---

> ### Author Rebuttal · Authors · 2026-03-29
>
> We thank the reviewer for recognizing the coupled PDE results, the KdV experiment, and the pipeline quality. It seems that there is a misunderstanding of the method and categorical errors about the tasks tackled in this work, which we hope to clarify below.
>
> **On circularity and "coefficient fitting."** These claims are factually incorrect.
> The grammar is not constructed per benchmark. SIGS uses a single fixed grammar of 51 production rules across all ten benchmarks without modification (L246–249). Wave eigenfunctions, diffusion modes, shock profiles, Gaussians, polynomials, transport phases, and random compositions all coexist (Tables 6–7 span 14 PDE families). The search must select the correct family, instance, and composition from this shared pool — not fill in coefficients of a pre-identified template.
> The method is not a coefficient fitter. The reviewer's reading appears to assume a SINDy-like workflow: select a few atoms, place them in slots, fit coefficients. That is not what SIGS does. The grammar generates a corpus of ~24,000 symbolic compositions (Table 11). Stage I must determine which of these compositions fill each Ansatz slot — a combinatorial selection problem over the latent manifold, not regression over a pre-selected basis. Only after Stage I identifies the symbolic structure does Stage II optimize coefficients (Fig. 1, App. B). The Protocol 1 comparison confirms this distinction: HD-TLGP given the same atoms produces 35–178% error (Table 3); SSDE with tailored dictionaries produces 45–5870%. SIGS outputs compact expressions; SSDE produces 500+ operation trees; HD-TLGP nests sin(sin(cos(exp(···)))) (Tables 19–21).
> The ablation is also misread. Experiment vi shows atoms are necessary; Protocol 1 shows they are not sufficient. A new experiment tests this further: we ran Burgers with K sum terms where the true solution requires K=1. For K∈{1,2,3}, machine precision is achieved (rel. L²≈10⁻¹³) with redundant coefficients driven to zero. At K=5, error remains 4.52×10⁻⁵. A coefficient fitter would not degrade gracefully under over-specification; SIGS does because Stage I performs genuine structure selection.
>
> **On removing entire atom classes.** If a complete atom class were absent and no equivalent in-grammar surrogate existed, any approximation method would fail — this is a fundamental limitation of approximation theory, not a SIGS-specific defect. No method (FEM, PINNs, or symbolic) can represent solutions outside its function class. KdV tests the meaningful version: the natural primitive is removed, an equivalence exists, and SIGS discovers it autonomously.
>
> **On SR baselines (PySR, LLM-SR, deep SR).** The suggested baselines are already represented. HD-TLGP is genetic programming with PDE-residual fitness — structurally equivalent to PySR + PDE residual. SSDE is deep RL + RNN suggested trees with PDE-residual fitness — structurally equivalent to deep SR + PDE residual. Both fail (Tables 3, 5). LLM-SR would require constrained decoding to guarantee syntactic validity and remains brute-force over combinatorial space — the bottleneck SIGS addresses. Also, LLM-SR addresses discovery of ODEs and it is not trivial to adapt it to solution of PDEs.  However, LLM-based Ansatz generation combined with SIGS's structured search is a natural future direction, and it is in our future plans.
>
> **On noise and MDBench/PDEBench.** This is factually incorrect. SIGS takes the symbolic PDE operator as input (Eq. 1), not trajectory data — there are no measurements to corrupt. Candidates are scored by substituting into the governing equation; no data-ingestion step where noise enters. The relevant robustness axis is grammar misspecification, tested via KdV.
>
> **On Mathematica.** DSolve has binary outcomes: closed-form solution or failure with no output — no error to report. When it returns infinite series with denominators like 1.69×10⁶³ (Eq. 17), these are unusable.
>
> **On Ansatz effort.** SIGS requires less per-problem input than baselines: fixed grammar vs SSDE's per-problem dictionaries (L238–244) and HD-TLGP's problem-specific knowledge bases (App. C.3.2).  See the discussion with reviewer geLm for a comprehensive discussion on Ansatze.
>
> **On originality.** The TGVAE is ablated: 3.56% fewer degenerate decodes vs vanilla GVAE (L770–784). More broadly, this is not incremental engineering. SIGS converts a combinatorially intractable symbolic search into a tractable one via grammar atoms and global/local search on a continuous manifold. This method allows approximating the solution of coupled nonlinear PDE systems, which is not possible by any other method.
>
> **On benchmark scope.** 4/10 benchmarks go beyond known-solution recovery: PG-2/3/4 have no known closed form, KdV and CE test misspecification. The grammar's function class is not "inherently narrow": 51 rules generate compositions of trigonometric, exponential, polynomial, hyperbolic, and localized functions spanning 14 operator types (Tables 6–7).

---

> > ### Author Rebuttal · Reviewer_Efpo · 2026-03-31
> >
> > We thank the authors for their detailed rebuttal. Several points are well taken and we acknowledge them explicitly. However, we maintain our overall recommendation for the following reasons.
> >
> > ## Points we concede:
> >
> > The grammar circularity concern was overstated. We accept that SIGS uses a single fixed grammar across all ten benchmarks and that Stage I performs genuine structure selection over approximately 24,000 compositions. The over-specification experiment (K=1 to K=5) partially supports that the method does more than coefficient fitting, though we note this experiment was not in the original paper.
> >
> > The noise/MDBench point was a category error on our part. SIGS takes the symbolic PDE operator as input and scores candidates by substituting into the governing equation. There is no data ingestion step where measurement noise enters. We retract this criticism. The Mathematica comparison limitation is also fair: DSolve has binary outcomes and returns unusable infinite series, making quantitative comparison impractical.
> >
> > ## Points we maintain:
> > **On the fundamental completeness problem.** The most significant theoretical weakness is not addressed in the rebuttal. SIGS has no completeness guarantee: there is no theoretical assurance that the true solution lives within the grammar-constrained function space. PINNs are universal approximators with theoretical guarantees on approximation quality. FEM has convergence guarantees under mesh refinement. SIGS has no analogous property. The grammar is a finite hand-crafted function class with no density guarantee in any meaningful function space.
> >
> > This creates a practically dangerous failure mode. When SIGS returns a symbolic expression with non-zero residual on a new problem, there is no way to distinguish between "the method has not converged" and "the true solution is not expressible in this grammar at all." The KdV experiment tests one convenient case where a known algebraic equivalence exists, which is insufficient to support general robustness claims.
> >
> > **On the optimization landscape.** PINNs already face well-documented difficulties with residual minimization: collocation point pathologies, loss balancing, and spectral bias. SIGS performs essentially the same residual minimization over a far more restricted and non-differentiable search space. The success of SIGS therefore depends entirely on the grammar containing the correct function family, returning to the completeness concern above.
> >
> > **On the proper benchmark framework.** The correct benchmark for a pure equation-driven solver is PINNacle (Hao et al., 2023), specifically designed for this setting with standardized evaluation protocols. Its absence is not justified. Furthermore, since SIGS scores candidates purely via PDE residual, augmenting this with a data term is straightforward, making PDEBench and MDBench runs feasible for a data-augmented extension the paper does not acknowledge.
> >
> > **On modern symbolic regression baselines.** The claim that HD-TLGP is structurally equivalent to PySR with PDE residual fitness is asserted without justification. PySR incorporates substantially more sophisticated search strategies. The dismissal of LLM-SR as "brute force" mischaracterizes the method and the authors acknowledge they have not tested it. These baselines remain absent.
> >
> > **On the TGVAE contribution.** A 3.56% reduction in degenerate decodes is thin evidence for a major architectural contribution.
> >
> > **On reproducibility.** The Ansatz design effort comparison deflects rather than addresses the concern. The question is whether a practitioner could apply SIGS to a genuinely novel PDE without significant domain expertise. The paper provides insufficient documentation to answer affirmatively.
> >
> > **Summary.**
> >
> > The rebuttal partially addresses the grammar circularity concern and correctly identifies the noise comparison as a category error. However the fundamental completeness gap, missing PINNacle comparison, unconvincing dismissal of modern SR baselines, and thin TGVAE ablation remain unaddressed. We maintain our recommendation of rejection with encouragement to resubmit with these issues addressed.

---

> > > ### Author Response · Authors · 2026-04-01
> > >
> > > The reviewer’s original assessment rested on two claims: the evaluation is circular, and noise robustness is untested. Both have now been explicitly addressed and conceded as such by the reviewer. The new arguments — completeness guarantees, PINNacle, PySR, LLM-SR — share a common pattern: they judge a structured symbolic discovery method by metrics and standards designed for numerical PDE solvers. This is an incorrect framing of our approach. SIGS is not competing with PINNs as a numerical PDE solver; it has a very different role in discovering compact symbolic expressions under explicit admissible priors — a task no other method can perform on coupled nonlinear PDE systems. Our work should be evaluated on this basis.
> > >
> > > We elaborate below.
> > >
> > > **On completeness — the new central argument.**
> > > This was not in the original review. The reviewer argues PINNs have universal approximation guarantees while SIGS does not. This is incorrect on both counts.
> > >
> > > First, the reviewer’s claim that the grammar is “a finite hand-crafted function class with no density guarantee” ignores the fact that the compositions of elementary functions (polynomials, trigonometric, exponential) are dense in C(K) for compact K by Stone-Weierstrass. The grammar’s function class *is* a universal approximator in the limit of unbounded expression length, just as neural networks are in the limit of unbounded width. The practical constraint for both is finite complexity — finite expression depth for SIGS, finite network width for PINNs. The expressivity concern is symmetric, not one-sided.
> > >
> > > Second, the reviewer conflates expressivity with optimization. PINNs have universality but achieve 2.56–6.09% error on these benchmarks and fail on coupled systems. SIGS achieves 10⁻¹³ error on the same problems. A theoretical guarantee that the optimizer cannot realize is not operationally meaningful. The “dangerous failure mode” (cannot distinguish non-convergence from inexpressibility) applies equally: when a PINN returns 6% error, one cannot tell if it is architecture limitation or optimization failure.
> > >
> > > **On the optimization landscape.**
> > > The reviewer claims SIGS performs “residual minimization over a non-differentiable search space.” This is factually wrong. Stage I searches a continuous latent manifold (clustering and sampling); Stage II optimizes a handful of scalar coefficients on a smooth landscape. PINNs optimize millions of weights with documented pathologies. These are qualitatively different problems.
> > >
> > > **On PINNacle.**
> > > PINNacle benchmarks neural PDE solvers using grid-based field accuracy — evaluation protocols for methods that output numbers on grids. SIGS outputs symbolic expressions. The output types do not match. Moreover, PINNacle does not contain any symbolic method, making a fair comparison impossible.
> > >
> > > **On PySR.**
> > > We concede “structurally equivalent” was too strong. But Virgolin & Pissis (2022, already cited in our paper) prove symbolic regression is NP-hard over the function space defined by primitive combinations — the space PySR searches. More sophisticated heuristics for an NP-hard discrete problem do not change the complexity class. SIGS addresses this by embedding grammar-generated compositions into a continuous latent manifold, it replaces combinatorial tree enumeration with geometric search. . PySR has never been demonstrated on equation-driven coupled nonlinear PDE systems.
> > >
> > > **On LLM-SR.**
> > > LLM-SR is data-driven, fitting expressions to observed trajectories. Adapting it to equation-driven PDE discovery requires constrained decoding, a PDE-residual backend, coupled multi-field handling, and Ansatz integration, leading to a new research paper, rather than just a missing baseline.
> > >
> > > **On TGVAE.**
> > > We did not claim TGVAE as a standalone contribution of this paper. It is a novel feature that adds to improving the performance that we have clearly demonstrated.
> > >
> > > **On Ansatz effort.**
> > > Every method requires comparable domain input. FEM: mesh and basis functions. PINNs: architecture, activations, collocation, loss weights. SIGS: Ansatz. Claiming SIGS’s documentation is insufficient while accepting PINNs — which require extensive per-problem tuning is unfair, in our opinion.

---

### Official Review · Reviewer_gAZC · 2026-03-08

**Soundness:** 3
**Presentation:** 1
**Significance:** 3
**Originality:** 3
**Overall Recommendation:** 4
**Confidence:** 2

**Summary:**

This paper introduces the Symbolic Iterative Grammar Solver (SIGS), a grammar guided neuro-symbolic framework for discovering analytical solutions to PDEs. The method constructs candidate solution atoms from a CFG, composes them through a human desgined Ansatz. Expressions are embedded into a continuous latent space using TGVAE (Topological Grammar VAE), and searches this space using a residual loss derived from the governing equation and initial, boundary conditions. The empirical study covers canonical PDEs with known analytical solutions, PDEs without known closed-form solutions, a grammar-misspecification setting, and coupled nonlinear systems, with the goal of recovering either exact symbolic solutions or accurate analytical approximations.

**Compliance With Llm Reviewing Policy:**

Affirmed.

**Final Justification:**

The paper presents a technically coherent and promising neuro-symbolic approach for symbolic PDE solution discovery, with interpretable outputs and strong results on the selected benchmarks. My main reservation remains that the method relies heavily on structured human priors through the Ansatz, grammar, and atom library, and I also found the empirical support for misspecification robustness and unknown-solution approximation somewhat limited. Still, I find the contribution sufficiently interesting and well executed to support a weak accept, and the rebuttal did not change my overall evaluation, so I am maintaining my original score.

**Key Questions For Authors:**

In addition to the concerns raised in the Weaknesses, I have several minor questions and requests for clarification:

1. What exactly is meant by "exact solution" in Table 1? Even if the discrete functional form is exactly matched, it is much less obvious how the coefficients can be identified exactly through numerical optimization.

2. In the experiment of grammar misspecification, the grammar lacks $cosh$ and $sech$, but the recovered alternative is essentially an equivalent reparameterization via $tanh$. Could the authors comment on whether the method can still recover meaningful approximations under more substantive misspecification, rather than identities of the form $sech^2(x)=1-tanh^2(x)$?

3. How should readers interpret the results for PDEs without known analytical solutions: as evidence of open-ended symbolic discovery, or as evidence that SIGS can fit accurate symbolic approximations within a specific domain?

4. In the discussion, what exactly is meant by the "one-time pretraining step"? Is this referring only to the GVAE manifold construction, or to a broader pretraining stage?

**Limitations:**

yes

**Strengths And Weaknesses:**

## Strengths

- **Technically coherent neuro-symbolic design:** The combination of grammar-constrained expression generation, latent-space search, and residual-based refinement is well motivated. The hierarchical use of atoms and Ansatz is a reasonable way to reduce combinatorial complexity relative to flat primitive-level search.

- **Strong empirical results on the chosen benchmark suite:** On the benchmark problems with known solutions, SIGS achieves extremely small relative errors, and the experiments on coupled nonlinear systems and grammar misspecification are interesting extensions beyond the most standard symbolic-discovery setting.

- **Interpretable outputs:** A meaningful strength of the work is that the method returns compact symbolic expressions rather than only black-box numerical approximations. This is valuable for scientific computing settings where explicit structure matters.

- **Potentially useful direction for PDE-oriented symbolic discovery:** Within the narrower problem setting of analytical PDE solution discovery under structured priors, the paper offers a contribution that others are likely to build on.


## Weaknesses
- **Related work and novelty are positioned too narrowly around recent PDE-specific baselines.** The paper discusses SSDE and HD-TLGP, but it does not adequately situate SIGS within the broader symbolic-regression literature. In particular, foundational or highly relevant references such as Schmidt and Lipson[1], AI Feynman[2], Biggio et al.[3] are missing, even though SIGS also combines symbolic structure, neural guidance, and strong search priors. As written, the manuscript overstates how fundamentally new the overall paradigm is.

- **The method depends heavily on human priors, which should be acknowledged more directly.** The main source of tractability seems to originate from the problem-specific Ansatz, the grammar design, and the atom library. These are strong forms of expert knowledge that substantially restrict the search space. This is not necessarily a flaw, but it creates tension with the framing of SIGS as an *automated analytical solution discovery method*.

- **The ablation evidence is incomplete relative to the paper's claims.** The manuscript argues that SIGS succeeds because of the combination of grammar-constrained atoms, latent-manifold search, and the overall hierarchical design. However, the ablation mainly supports atoms over primitives. It does not isolate the effect of the topological regularization, the iterative clustering/search procedure, or the strength of the manually chosen Ansatz. As a result, it remains unclear which components are essential and which are secondary.

- **The "unknown-solution" experiment is still strongly structured by human design.** For the Poisson-Gauss problems, the exact closed form is unknown, but the experiment still uses a tailored Ansatz based on Gaussian components, and even varies the number of modes with the instance. This makes the result interesting as a structured symbolic approximation experiment, but less convincing as evidence of open-ended discovery without strong prior knowledge. Moreover, once the target problem is framed as approximation without analytical ground truth, comparison against numerical or neural approximation methods such as PINNs or other neural/numerical solvers would be more appropriate for assessing practical accuracy.

- **Presentation clarity can be improved.** Parts of the method section are written in an overly abstract style, with notation and mappings that obscure the main idea (e.g., $\tau$, $\kappa$ mapping). The solution-discovery section in particular would benefit from a more direct algorithmic explanation. Some smaller presentation issues also remain, such as notation inconsistencies around Fig. 1 and its caption ($s \in \mathcal{L}(\mathcal{G})$ vs  $w \in \mathcal{L}(\mathcal{G})$)

---

> ### Author Rebuttal · Authors · 2026-03-29
>
> We thank the reviewer for recognizing the technical coherence of the design, the strength of the coupled-system and misspecification results, the value of interpretable outputs, and the potential for others to build on this direction. We also appreciate the notational catches around Fig. 1. We address each weakness and question below.
>
> **Related work positioning.** The reviewer is right that the broader SR literature should be acknowledged more explicitly. We believe the intended references are Schmidt & Lipson (2009, Science), Udrescu & Tegmark (2020, Science Advances) on AI Feynman, and Biggio et al. (2021, PMLR). We will cite these and situate SIGS relative to them. The key distinction we will make precise: these methods are data-driven, fitting symbolic expressions to observed trajectories. SIGS is equation-driven, minimizing the PDE residual directly without numerical data (Eq. 2). Our novelty claims are specific to the equation-driven PDE setting, where no prior method combines grammar-constrained generation with latent-space search. We will revise the related work to make this boundary explicit rather than implicit.
>
> **Human priors.** We agree the role of the Ansatz should be acknowledged more directly. SIGS automates the search given a practitioner-specified Ansatz, analogous to how FEM automates the solve given a practitioner-specified mesh and basis (L146–164). The grammar is fixed: 51 rules, 14 PDE families, shared across all benchmarks (L246–249). Only the high-level Ansatz adapts. Importantly, SIGS requires less per-problem input than baselines: SSDE needs per-problem tailored dictionaries (L238–244) and HD-TLGP needs problem-specific knowledge bases (App. C.3.2). The reviewer can also read the first paragraph of the response to reviewer geLm for further discussion on Ansätze.  We will reframe the contribution as structured analytical discovery under admissible priors in revision.
>
> **Ablation completeness.**  In addition to the referred experiments we provide evidence across experiments that support both the role of the regularization, atoms and primitives : TGVAE requires 3.56% fewer decode attempts than vanilla GVAE (L770–784); HD-TLGP Protocol 1 with the same atoms produces 35–178% error (Table 3), confirming atoms are necessary but not sufficient — the GVAE-based search bridges this gap; and PG demonstrates operation under an underspecified Ansatz. We agree that these are distributed across experiments rather than presented as a unified ablation, and will consolidate this evidence more explicitly.
>
> **Poisson–Gauss interpretation.** We agree PG should be read as structured symbolic approximation, not open-ended discovery. One factual clarification: the PG Ansatz is u(x,y)=∑ⱼφⱼ(x,y)ψⱼ(x,y) (L296) — a generic sum of products of two functions, not a Gaussian-specific template. SIGS discovered that Gaussian-like atoms fit the problem through latent search; the Ansatz prescribes only the superposition structure. The mode count varies because more sources require more terms, as in FEM where the basis count is also user-chosen. Regarding comparison with PINNs/neural solvers: the accuracy reference is already present through FEniCS (Table 4, Table 26). SIGS's contribution for PG is providing what FEniCS cannot: a readable expression from which a practitioner directly reads centers, widths, amplitudes, and boundary compatibility (Table 19). This interpretability is the core motivation (L1–15).
>
> **Presentation.** We acknowledge that the search part is condensed due to page limitations. The algorithmic details are given in Appendix B (Algorithms 1–4) but should be surfaced, at least partially, in the main text. We will revise the method section: replace the τ and κ mappings with a explicit description, simplify notation around the assembly map A, and fix the Fig. 1 inconsistencies.
>
> **Q1:** Exact solution means the discovered expression matches the analytical ground truth up to float64 machine precision: relative L² errors of 10⁻¹³–10⁻¹⁴ (Table 3). Constants like π are grammar terminals (L594), not numerically approximated — unlike baselines which use e.g. 3.141 (Table 14). We will define this precisely in revision.
>
> **Q2/3:**  It depends. If the true solution were genuinely outside the approximation class with no equivalent in-grammar surrogate, SIGS would fail — but this is a fundamental limitation of approximation theory, not a SIGS-specific defect. No method (FEM, PINNs, or symbolic) can represent solutions outside its function class. We perform a new experiment for the Burgers equation under different levels of misspecification, and show robustness (see answer to Reviewer geLm).
>
> **Q4:** "One-time pretraining" refers solely to GVAE manifold construction (23,682 sequences, Table 11), that is constructed once and used for all downstream problems without retraining. We will make this explicit.

---

> > ### Author Rebuttal · Reviewer_gAZC · 2026-04-03
> >
> > Thank you for the detailed response. I appreciate the authors' clarifications regarding the Poisson-Gauss setting and the role of the Ansatz and other priors.
> >
> > That said, my main concern remains. The practical success of SIGS still appears to depend strongly on human priors through the Ansatz, grammar, and atom library, and this remains central to my evaluation. Relatedly, I do not find the grammar-misspecification evidence fully convincing. Although `sech`/`cosh` are removed, `tanh` is still available, and the KdV example is then recovered through an equivalent identity. In my view, this is not a strong demonstration of meaningful misspecification robustness. A more convincing case would be one where the target function is genuinely absent from the grammar, yet the method still recovers a useful approximation from a qualitatively different function family, for example a truncated Taylor expansion.
> >
> > I also find that the unknown-solution task (PG problem) would benefit from a neural baseline. While direct comparison to PDE solvers is not always straightforward in symbolic discovery, this experiment is already evaluated as an approximation problem relative to a FEniCS ground truth, so such a baseline would still be important for assessing practical performance in this particular downstream task.
> >
> > Overall, the rebuttal clarified several points and improved the calibration of the claims, but the remaining issues concern core aspects of the paper's positioning and empirical support. Accordingly, my current evaluation remains unchanged.

---

> > > ### Author Response · Authors · 2026-04-03
> > >
> > > We thank the reviewer for the continued engagement.
> > >
> > > **On human priors.**
> > > Every approximation method operates under human priors: FEM requires a mesh and basis, PINNs require an architecture and collocation strategy, separation of variables requires a coordinate system. The Ansatz is SIGS's version of this, and the paper says so explicitly (L146–164). Within this regime, SIGS requires strictly less per-problem input than the baselines: one fixed grammar across all benchmarks vs SSDE's per-problem dictionaries and HD-TLGP's problem-specific knowledge bases. The Burgers over-specification experiment , see response to Reviewer efpo, further shows the method is not brittle, we get machine precision for $K \leq 3$ graceful degradation at $K=5$,  which is inconsistent with a method that only works when the Ansatz is precisely matched to the answer.
> > >
> > > **On KdV and misspecification — the experiment the reviewer asks for already exists.**
> > > The reviewer asks for a case where the target is genuinely absent from the grammar and the method recovers a useful approximation from a qualitatively different family. This is exactly the Poisson–Gauss experiment.
> > >
> > > In PG, the true solution is a non-closed-form function arising from Gaussian forcing. It does not belong to any natural atom class in the grammar — it is not an eigenfunction, a shock profile, a transport wave, or any designed template. SIGS approximates it at 1–3% error (Table 4) by assembling available building blocks into a representation that has no structural relationship to the unknown ground truth. This is not in-grammar equivalence (KdV); it is out-of-family symbolic approximation; precisely what the reviewer is requesting.
> > >
> > > KdV and PG together cover two complementary misspecification regimes: KdV shows that when a natural primitive is removed but an equivalent exists, SIGS discovers it autonomously. PG shows that when no exact representation exists at all, SIGS produces accurate approximations from its symbolic vocabulary. We will connect these more explicitly in revision, but the evidence is already in the paper.
> > >
> > > **On a PINNs baseline for PG.**
> > > We can include PINNs accuracy on PG in revision. However, the purpose of the PG experiment is not to compete on field accuracy; FEniCS already serves as validated ground truth. The contribution is that SIGS returns a readable and **easily interpretable** symbolic expression (Table 19) from which a practitioner directly reads Gaussian centers, widths, amplitudes, and boundary compatibility. FEniCS returns numbers on a (fine) grid; PINNs will return network weights for a large-sized neural networ. Neither provides any structural insight in the manner that SIGS does. Adding a PINNs number contextualizes the error, but it does not change what the experiment demonstrates.
> > >
> > > **Closing.**
> > > The reviewer's evaluation correctly identifies SIGS as a structured discovery method under admissible priors. That is the paper's intended scope, and within that scope, SIGS achieves results no prior symbolic method can; including coupled nonlinear PDE systems (SWE, CE) where all baselines fail entirely. The concerns about Ansatz dependence and misspecification robustness are real scope boundaries, but the paper already contains stronger evidence for both than the review currently credits.

---

### Official Review · Reviewer_geLm · 2026-03-09

**Soundness:** 2
**Presentation:** 2
**Significance:** 2
**Originality:** 3
**Overall Recommendation:** 3
**Confidence:** 5

**Summary:**

This paper proposes SIGS, a framework for discovering analytical solutions to differential equations by performing search on a continuous latent manifold. The method represents candidate solution expressions using a context-free grammar (CFG) and trains a GVAE on a grammar-generated library of symbolic components (atoms and their compositions), spanning multiple differential-equation families, with geometry regularization to improve latent-space search. Given a target differential equation together with its boundary/initial conditions, SIGS then explores the corresponding region of the latent space, evaluates decoded candidates using a physics-based residual, and further refines the discovered expression by optimizing its numerical constants.

Empirically, SIGS evaluates on a benchmark suite spanning hyperbolic/parabolic/elliptic PDEs, including several scalar PDEs with known analytical solutions, Poisson problems with Gaussian source terms (treated as lacking a known closed form and evaluated against validated numerical references), a robustness test under grammar misspecification (KdV), and coupled nonlinear PDE systems (e.g., Shallow Water and Compressible Euler) via manufactured-solution setups. The paper positions SIGS as a unified approach that combines grammar-constrained symbolic composition with continuous latent optimization to recover exact solutions when they exist and to produce accurate, interpretable symbolic approximations otherwise, while using a single fixed grammar across tasks and adapting only the high-level ansatz per problem.

**Compliance With Llm Reviewing Policy:**

Affirmed.

**Final Justification:**

Thank you for the rebuttal. It usefully clarifies several points, including the distinction between corpus-construction-time filtering and search-time search, the intended interpretation of the coupled-system results, and the narrower scope of the misspecification claim. The authors strive to assess the context, and Overall, the authors investigate an important concept with several empirically strong results, especially on challenging PDE benchmarks. However, the rebuttal does not resolve my central reason for the original score: I still do not find the paper sufficiently sound in attributing its gains specifically to the latent-manifold search, as opposed to the combination of manually specified grammar design, admissibility constraints, and problem-dependent structural Ansätze that substantially restrict the search space. The response provides intuition for why the manifold should help, but it does not provide a clean isolating ablation that disentangles the contribution of latent search from these other restrictions. Likewise, the additional evidence on overspecification is useful, but it does not fully address the more important question of robustness under genuinely weak or partially misspecified priors. I also remain somewhat unconvinced on generality and originality: while the paper shows promising performance within the proposed setup, the effective dependence on human-designed priors and component libraries appears more central to success than the current framing suggests. Thus, although the rebuttal improves clarity and slightly strengthens the presentation, it does not materially increase my confidence in the method’s soundness or broader generality, so I maintain my original recommendation.

**Key Questions For Authors:**

1. How sensitive is SIGS to the manually specified Ansatz?

The method assumes that the target solution is exactly or sufficiently well represented by the chosen Ansatz, which also determines the admissible atom families/couplings. Can the authors provide evidence that performance remains strong under weaker or partially misspecified Ansatz choices? This would clarify whether the main contribution is the latent-manifold search itself or the manual restriction of the search space.

2. Can the authors clarify how the GVAE/TGVAE training corpus and typed CFG are constructed in practice?

It is still unclear how much of the component library comes from hand-designed atoms vs random compositions, what filtering/deduplication is applied, and how the typed CFG is designed. These are core parts of the method, not minor implementation details, and they affect both reproducibility and generality.

3. How should the coupled nonlinear system results (e.g., SWE/CE) be interpreted: exact symbolic recovery or high-accuracy approximation?

These experiments use manufactured analytical baselines, but the optimized parameters do not always appear to match the manufactured coefficients exactly. Please clarify the success criterion used in these cases. A consistent notion of “recovery” across scalar PDEs and coupled systems would strengthen the empirical claims.

4. Does the KdV misspecification experiment demonstrate transfer beyond the prebuilt library, or only recovery of an equivalent in-grammar expression?

My reading is that this result shows robustness when a natural primitive is missing, but an equivalent grammar-compatible form still exists. Do the authors have evidence for transfer when the true solution structure is genuinely outside the assumed grammar/Ansatz family?

5. Can the authors disentangle the effects of CFG validity, typed/tag-based latent-subspace restriction, and post-decode mathematical checks?

All three mechanisms reduce the search space, but the paper does not clearly show which one is most responsible for the reported gains. An ablation removing or relaxing typed constraints and mathematical checks would be very helpful. This is central to my soundness assessment.

**Limitations:**

yes

**Strengths And Weaknesses:**

**Soundness**

The overall pipeline  is plausible and appears technically coherent. The benchmark suite is reasonably broad (scalar PDEs, Poisson–Gauss with FEM reference, grammar misspecification, and manufactured coupled systems).
However, my main concern is that the paper’s strongest empirical results appear to depend heavily on a manually specified structural Ansatz. Since the method explicitly assumes that the target solution is exactly or approximately contained in this Ansatz family, the effective search problem may be much easier than generic symbolic PDE solving. The paper does not convincingly disentangle the benefit of the latent-manifold search from the benefit of restricting the hypothesis class through hand-designed admissibility constraints. The empirical claims are hard to interpret cleanly because performance seems to rely on multiple layers of engineered priors beyond the latent-manifold idea: (i) a hand-designed component/atom library, (ii) semantic-tag/type constraints restricting the searched latent region, and (iii) additional “mathematical checks” filtering candidates before residual evaluation. These choices materially reduce search difficulty, yet the paper does not provide convincing ablations to isolate what actually drives success.
For the coupled nonlinear systems (SWE/CE), the setups use manufactured analytical baselines, but results appear closer to “recovering correct structural modes with nontrivial coefficient deviations” than strict exact recovery; the paper should explicitly clarify whether these are evaluated as exact symbolic recovery (up to equivalence) or approximation, and use consistent success criteria.

**Presentation**

The paper is readable and the high-level narrative is clear, but several details remain underspecified for an expert reader to fully reproduce the method. In particular, the construction of the GVAE training corpus is not sufficiently transparent: it is still unclear how the component library is assembled in practice, how many samples come from hand-designed atoms versus random compositions, what filtering and deduplication steps are applied, and how sensitive the method is to these choices. Likewise, the paper states that a formal/typed CFG is used, but the process by which this grammar is actually designed remains insufficiently explicit. Since the grammar and the training corpus jointly define the hypothesis class explored by the latent search, these are not minor implementation details but central parts of the method. The paper would benefit from a much clearer description of (i) library composition, (ii) typed/tagging rules, (iii) exact filtering checks, and (iv) search and refinement hyperparameters. More generally, the boundary between CFG validity, typed filtering, and mathematical checks should be stated explicitly, ideally with a simple schematic.

**Significance**

Symbolic PDE solving is an important problem, and the continuous-manifold search perspective is potentially useful. The KdV experiment provides some evidence of robustness to moderate grammar misspecification, since SIGS can recover an equivalent form when a single natural primitive is missing. However, this does not fully resolve the broader concern of transfer beyond the prebuilt library and admissibility machinery. In KdV, the target solution still appears to admit a grammar-compatible equivalent representation, and the search remains guided by a hand-specified Ansatz. More generally, the paper itself acknowledges that the framework depends on the joint design of grammar, Ansatz, and latent space, and that human expert choices still play an important role. As a result, it remains unclear how well the method would transfer to PDE families whose true solution structures are genuinely outside the assumed atom/Ansatz family, rather than merely expressible through an equivalent in-grammar surrogate. The practical impact would be much stronger with evidence of transfer under minimal re-engineering.

**Originality**

The work is a nontrivial combination of known ideas (grammar-constrained SR, VAE latent search, residual-based scoring, constant refinement, plus topology/geometry regularization). The integration is reasonably novel as an end-to-end system, but the novelty attributable specifically to the latent-manifold search (as opposed to strong priors/filters) is not well demonstrated.

---

> ### Author Rebuttal · Authors · 2026-03-29
>
> We thank the reviewer for the technically precise reading. We agree the main text should separate the roles of grammar, corpus, admissibility, and search more explicitly. We correct several factual points below.
>
> **Ansatz and search difficulty.** An Ansatz is not a SIGS-specific weakness — FEM, neural networks, separation of variables, HD-TLGP, and SSDE all impose a function class before optimization (L146–164). The question is how restrictive the prior is. SIGS reuses a fixed grammar across all benchmarks (L246–249), while SSDE needs per-problem dictionaries (L238–244) and HD-TLGP needs problem-specific knowledge bases (App. C.3.2). This matters differently for exact recovery and approximation. For exact recovery, the Ansatz can be limiting — we agree to this. For approximation, broad superposition families suffice. In PG, no solution is known and SIGS uses only a generic superposition u(x,y)=∑ⱼφⱼ(x,y)ψⱼ(x,y), achieving 1–3% error (Table 4). In CE, the Ansatz is overcomplete: 24 atom slots across four coupled fields, making optimization harder, not easier. CE is accordingly the hardest benchmark (~10% error).To directly test Ansatz sensitivity, we deliberately over-specified the Burgers Ansatz with K sum terms where the true solution requires K=1 — making the optimization landscape strictly harder by introducing redundant degrees of freedom. For K∈{1,2,3}, machine precision is achieved (rel. L²≈10⁻¹³) with redundant coefficients driven to zero. At K=5, error is still 4.52×10⁻⁵. This answers both Q1 and Q5: SIGS is robust to moderate over-specification, and the search/optimization is doing genuine structure selection under an enlarged hypothesis class, not merely fitting coefficients inside a minimal template.
>
> **Disentangling priors from the latent search.** The review's claim that "additional mathematical checks filter candidates before residual evaluation" is inaccurate. The checks in App. A.1.3 (canonization, deduplication) occur only during corpus generation. During search, the sole restriction is variable/tag admissibility; candidates are scored directly by R(u) with no intermediate expert filtering. Tags are simple arity labels excluding incompatible components, e.g. considering 1D atoms for 2D PDEs, not hidden answer-specific biases. Evidence on three axes: atoms necessary (exp. vi: 133/50,000 admissible, best ~366%); atoms not sufficient (HD-TLGP Protocol 1 same atoms: 35–178%; SSDE tailored: 45–5870%); topology helps (TGVAE: 3.56% fewer degenerate decodes, L770–784). We agree, however, that these experiments are a bit scattered and will fix this.
>
> **Originality of the latent-manifold search.** The reviewer asks what the latent search contributes beyond strong priors. The answer is geometric structure over the space of admissible symbolic compositions.
> Embedding atoms into a continuous manifold gives the search: (i) a metric where nearby vectors decode to similar expressions; (ii) density-based clustering over promising families; (iii) smooth traversal without discontinuous tree mutation.
> Discrete search, even with identical atoms, must enumerate combinatorially with no functional-similarity metric, and no way to allocate samples to promising regions. HD-TLGP Protocol 1: same atoms, 35–178% error, 100–1000× slower (Table 5). The priors define what is searched; the manifold defines how.
>
> **Coupled-system success criteria.** The intended interpretation is tiered: scalar PDEs → exact recovery at 10⁻¹³–10⁻¹⁴ (Table 3); SWE → high-accuracy, per-field <0.05% (Table 22); CE → approximation at ~10% (L381). We will state these explicitly.
>
> **KdV interpretation.** We agree that KdV is not unrestricted out-of-family transfer. But the equivalence was not given: sech/cosh were removed (L363–366), and SIGS found 2−2tanh²(x−4t) purely via residual minimization with no knowledge of sech²=1−tanh². The Ansatz is a generic polynomial in one atom up to degree 2 (Table 2). If the true solution were genuinely outside the approximation class with no equivalent in-grammar surrogate, SIGS would fail — but this is a fundamental limitation of approximation theory, not a SIGS-specific defect. No method (FEM, PINNs, or symbolic) can represent solutions outside its function class.
>
> **Corpus and grammar.** The corpus (23,682 sequences, Table 11) is generated once from the fixed grammar: the code randomly mixes operator-motivated templates and random compositions (App. A.1), and we do not control this ratio — we generate once and use whatever the pipeline produces. To clarify a possible misreading: the grammar is a plain CFG, not a typed or attribute grammar. "Types" in App. A.2 are post-hoc variable-presence tags computed from decoded parse trees, used only downstream to define admissible search subspaces — not part of the grammar formalism. For this reason, it is not well-defined to define a sensitivity analysis over corpus generation as it is done automatically.

---

> > ### Author Rebuttal · Reviewer_geLm · 2026-04-03
> >
> > Thank you for the detailed rebuttal. I appreciate the clarifications, especially the distinction between the fixed CFG, the post-hoc variable/tag admissibility mechanism, and the fact that canonization/deduplication are applied during corpus generation rather than during search. The rebuttal also helps clarify the intended interpretation of the coupled-system results: scalar PDEs are positioned as exact recovery cases, while SWE and CE should be read as high-accuracy approximation results rather than uniform exact symbolic recovery. I also appreciate the authors’ explicit acknowledgment that the KdV misspecification result does not demonstrate unrestricted out-of-family transfer.
> >
> > That said, the rebuttal does not sufficiently change my overall assessment. My main concern was whether the empirical gains can be clearly attributed to the latent-manifold search itself, rather than to manually imposed restrictions on the search space. On this point, I do not think the response fully resolves the issue. The new evidence on Burgers mainly addresses robustness to overspecification, which is useful, but it does not directly answer the more central question of robustness under partial misspecification or genuinely weak Ansatz choices. Similarly, while the rebuttal clarifies that some of the filtering/checking mechanisms operate only at corpus-construction time, it still does not provide a clean ablation disentangling the respective contributions of CFG validity, tag/admissibility-based latent-subspace restriction, and other search-space constraints.
> >
> > I also still have some reservations regarding reproducibility and generality. The additional explanation of how the corpus is generated is helpful, but the practical construction of the component library, the generation process, and the effective dependence on manually specified priors remain somewhat underspecified relative to their importance in the method. More broadly, the rebuttal improves the calibration of several claims, but in my view it does not add enough new evidence to materially strengthen the paper’s soundness or originality.
> >
> > For these reasons, although I appreciate the clarifications and think they improve the presentation of the work, they do not sufficiently address my central concerns to justify raising my score.

---

> > > ### Author Response · Authors · 2026-04-04
> > >
> > > We thank the reviewer for acknowledging the clarifications on corpus-time mechanisms, coupled-system interpretation, and KdV scope.
> > >
> > > **On reproducibility.** We would like to emphasize that the corpus construction did not require any special tuning that might affect Reproducibility. In fact, the generation pipeline randomly mixes operator-motivated templates and random compositions with no controlled ratio; we did not optimize what to include, how much of each family, or any balancing parameter. We generated one corpus and used whatever the pipeline produced across all ten benchmarks. There is no tuning whatsoever. The grammar is a plain CFG (51 rules), the corpus is 23,682 sequences, all Ansätze are in Table 2, hyperparameters in Table 10/App. B.
> > >
> > > **On “not enough new evidence to strengthen soundness or originality.”** We would like to reiterate that SIGS solves coupled nonlinear PDE systems (SWE, CE) where every prior symbolic method fails entirely. It recovers exact solutions at 10⁻¹³ where baselines with equivalent building blocks produce 35–5870% error. It approximates unknown solutions at 1–3% under completely uninformed Ansätze. Given these results and no specific asks from the reviewers, it is unclear to us what additional evidence is necessary to strengthen the paper.
> > >
> > > **On the role of the latent manifold:** In our opinion, the reviewer’s central question is what the latent search contributes beyond the manually imposed restrictions. To answer this question, we start by observing that the latent manifold imposes geometric structure on the space of admissible symbolic compositions. Discrete symbolic search has building blocks but no map: there is no notion of “nearby” parse trees that preserves functional similarity, no density structure to identify promising regions, and no smooth direction towards a lower residual. The latent manifold provides all three. Neighboring latent vectors decode to functionally similar expressions (metric structure), high-density regions correspond to coherent symbolic families that can be searched preferentially (clustering), and continuous traversal explores structural variations without the discontinuous jumps of tree mutation (smoothness).
> > >
> > > This is not one of several optional components — it is the mechanism that converts “having the right atoms” into “finding the right composition.” Every piece of evidence in the paper demonstrates this same principle:
> > >
> > > HD-TLGP Protocol 1 has the same atoms as SIGS but no manifold. It searches discrete trees by stochastic recombination. Result: 35–178% error, 2,000–14,000s. Same ingredients, no geometric structure → failure.
> > >
> > > SIGS with the same atoms and the manifold: 10⁻¹³ error, 9–111s. Same ingredients, geometric structure → success.
> > >
> > > Experiment vi removes atoms entirely: primitive sampling yields 133/50,000 admissible at ~366%. No ingredients, regardless of structure → failure.
> > >
> > > Burgers K=1→5 enlarges the search space by over-specifying: machine precision for K≤3, graceful degradation at K=5. The manifold navigates an enlarged hypothesis class — inconsistent with template fitting.
> > >
> > > These are not scattered observations. They are four views of one principle: the manifold is necessary because it is what makes search over admissible compositions tractable. The reviewer frames the missing piece as “a clean ablation disentangling CFG validity, tags, and search.” But the comparison that matters — same atoms with vs without the manifold — already exists as Protocol 1 vs SIGS. A unified table would be cleaner, and we will include one. But the evidential content is already present.
> > >
> > > **On genuinely weak Ansatz choices.** The reviewer distinguishes over-specification from genuine misspecification. PG is genuine misspecification: the true solution is non-closed-form, belongs to no atom class in the grammar, and the Ansatz is a generic superposition that does not encode the solution structure. SIGS achieves 1–3% by assembling building blocks into a representation qualitatively different from the ground truth. This is approximation under a structurally uninformed Ansatz — the scenario the reviewer requests.
> > > More broadly, asking SIGS to succeed under “genuinely weak” priors asks it to work without its core design principle. The Ansatz is not a crutch — it is the method, just as the mesh is FEM’s method and the architecture is PINNs’ method. The question is whether SIGS requires more restrictive priors than alternatives. It does not: one fixed grammar vs per-problem dictionaries (SSDE) and problem-specific knowledge bases (HD-TLGP).

---

### Official Review · Reviewer_UGH4 · 2026-03-11

**Soundness:** 3
**Presentation:** 2
**Significance:** 2
**Originality:** 3
**Overall Recommendation:** 4
**Confidence:** 3

**Summary:**

The paper proposes a neuro-symbolic framework for finding analytical solutions for partial differential equations (PDE). The approach employs a formal grammar (context-free grammars) for generating only syntactically valid solution building blocks. Additionally, the GVAE is employed for encoding the generated expressions in a continuous space.

**Compliance With Llm Reviewing Policy:**

Affirmed.

**Final Justification:**

my concern about expert Ansatz selection is partially resolved, I agree that the method seems to be still more roubst than the alternatives, but the limitation of setting up a correct Ansatz beforehand remains, i wll slightly adjust my score

**Key Questions For Authors:**

See the weaknesses above, in particular:

* Motivate the problem setting , why searching for analytic solutions of PDEs is a relevant problem for machine learning and AI methods  if very precise numerical solvers (FeniCS) and neural solvers (Fourier Neural Operator) exists, when the presented method will have clear advantages as compared to simply numerical solving the PDEs , and for what use-cases in particular it will be better ?

* Please clarify how can you verify numerically in practice the claim "SIGS achieves exact solutions on all problems, as it contains π as a symbol and does not approximate it numerically."

**Limitations:**

yes

**Strengths And Weaknesses:**

## Strengths

* The idea of generating solution building blocks using context-free grammars is interesting, sound, and appears novel.

* The method improves over existing state-of-the-art baselines. In the case of simpler 1D PDEs, the proposed approach is able to recover the exact analytical solution, whereas baseline methods typically obtain only an approximation.

* The paper includes a detailed empirical study, with numerous ablation experiments and an extensive evaluation of the proposed method.

## Weaknesses

* The method requires substantial domain knowledge in order to define an appropriate Ansatz for the problem under consideration so that solutions minimizing the physics-informed loss can be found. In my view, this is the main limitation of the approach. A crucial experiment appears to be missing that would evaluate the generalization capability of the method—for example, solving each equation starting from a general Ansatz (e.g., based on orthogonal basis functions), or identifying a generic set of grammar building blocks that would allow end users to apply the method to a new PDE without requiring significant domain expertise.

* I am also skeptical about the claim that vanilla ChatGPT is a viable tool for finding analytical solutions to PDEs in this setting. Recent agent-based or tool-augmented systems for mathematical reasoning may provide a more appropriate comparison. For example, the paper “Accelerating Scientific Research with Gemini: Case Studies and Common Techniques” discusses approaches designed for solving complex scientific and mathematical problems, which could serve as a more relevant baseline or point of comparison.

* A more general concern relates to the fundamental motivation for the proposed approach. If a PDE can already be solved numerically with high accuracy and efficiency using established numerical solvers (e.g., FEniCS), it would be helpful for the paper to better articulate why an ML/AI-based approach is preferable or necessary in this setting. Clarifying the scenarios in which the proposed method offers advantages over traditional numerical methods would strengthen the overall motivation of the work.

## General Assessment

* currently I'm voting borderline reject, but may update my score after reading the other reviews and the rebuttals.

---

> ### Author Rebuttal · Authors · 2026-03-29
>
> We thank the reviewer for recognizing the novelty of the grammar-based approach, the empirical gains over baselines, and the breadth of the study. We address the main concerns below.
>
> **The role of the Ansatz and domain knowledge**
> We agree that the high-level Ansatz is a limitation when the goal is exact symbolic recovery, since the true solution must lie in, or be equivalent to, the chosen symbolic family. The relevant comparative question is not whether SIGS uses an Ansatz, but how restrictive the required prior is compared to alternatives. Here, an important asymmetry exists: SIGS reuses the same fixed grammar and shared corpus across all benchmarks, while baselines like SSDE requires per-problem tailored dictionaries and HD-TLGP requires problem-specific prior knowledge. This matters differently for exact recovery and symbolic approximation. For exact recovery, the Ansatz can be limiting. For approximation, broad superposition families can still be useful. The paper gives two examples. In Poisson–Gauss, no analytical solution is known, and SIGS uses only the generic superposition \(u(x,y)=\sum_j \phi_j(x,y)\psi_j(x,y)\), achieving 1–3% relative \(L^2\) error against FEniCS. In Compressible Euler, the Ansatz is broad rather than narrowly matched: six atoms per field with exponential envelopes for \(\rho\) and \(p\), giving 24 atom slots across four coupled fields.  This is overcomplete rather than minimal, making optimization harder, not easier. CE is accordingly the hardest benchmark. So the correct scope is structured symbolic discovery under admissible priors, not prior-free open-ended discovery.
>
> **2. The requested generalization experiment**
> The paper already provides the reusable building-block layer the reviewer asks for. SIGS uses a single fixed grammar (51 production rules), a fixed atom-generation pipeline, and one shared GVAE/TGVAE manifold across all ten benchmarks. What changes per problem is the high-level assembly Ansatz, not the grammar or the learned symbolic manifold. We agree the paper does not consider a generic weak Ansatz on all problems. However, we provide an Ansatz ablation in the Response to Reviewer Efpo (§2) and show that SIGS achieves machine precision for levels of misspecified Ansatz. This is an encouraging first result of Ansatz robustness. We agree that automated Ansatz search is the natural next step in research.
>
> **3. ChatGPT and agent-based systems**
> We agree vanilla ChatGPT is not a task-matched baseline; it is included only as a human expert proxy. The results demonstrate its limitations: on PG-2 it returns an incorrect expression (157.6% relative \(L^2\) error, violating Dirichlet BCs); on Burgers it succeeds only by pattern-matching the tanh structure from the manufactured data rather than structured PDE solving. Richer agentic systems are relevant but not controlled baselines for grammar-constrained, residual-verified PDE solution discovery; a future direction is using an outer-loop AI to propose Ansätze while SIGS performs constrained symbolic search within the chosen family.
>
> **4. Relevance relative to numerical and neural solvers**
> Numerical solvers such as FEniCS return accurate values on a grid; neural operators such as FNO learn data-driven black-box maps between function spaces. SIGS instead evaluates symbolic candidates directly through the PDE residual and boundary/initial conditions, without per-problem simulation-data training, and returns compact expressions that expose mechanisms, symmetries, decay rates, wave speeds, and parameter dependence explicitly. The goal is not to replace these methods on raw accuracy, but to complement them where symbolic structure matters: interpretability, manufactured solutions, explicit parameter dependence, and understanding how a PDE solution is constructed.
>
> **5. On exactness, especially with $\pi$**
> We agree this should be stated more carefully. Symbols such as \(\pi\) are grammar terminals and therefore exact at the symbolic level; trainable constants are float64 and exact only up to numerical tolerance. "Exact solution" should mean exact functional recovery up to symbolic equivalence, with machine-precision verification on known-solution scalar cases — those achieve relative \(L^2\) errors of \(10^{-13}\)–\(10^{-14}\). The advantage of symbolic \(\pi\) is visible in the parity table, where HD-TLGP approximates \(\pi\) numerically while SIGS represents \(\sin(\pi x)\) exactly.
>
> **Summary**
>
> We agree the Ansatz is the bottleneck for exact symbolic recovery. But SIGS uses a fixed reusable grammar and atom library across all benchmarks, strictly less problem-specific than the baselines, and broad Ansätze already yield useful approximations when no exact solution is known. We will revise to sharpen the scope, clarify the motivation relative to numerical solvers, narrow the language around exactness, and state explicitly that ChatGPT serves only as difficulty calibration.

---

> > ### Author Rebuttal · Reviewer_UGH4 · 2026-04-03
> >
> > thanks for the rebuttal, my concern about expert Ansatz selection is partially resolved, I agree that the method seems to be still more roubst than the alternatives, but the limitation of setting up a correct Ansatz beforehand remains, i wll slightly adjust my score

---

> > > ### Author Response · Authors · 2026-04-04
> > >
> > > We thank the reviewer for the thoughtful follow-up and for the positive update. We appreciate their recognition that SIGS appears more robust and less restrictive than the alternatives studied.
> > >
> > > We agree that dependence on a practitioner-specified high-level Ansatz remains a limitation, especially for exact symbolic recovery. This dependence is also not unique to SIGS, since practical methods such as FEM, PINNs, and symbolic approaches all rely on a chosen approximation space.
> > >
> > >  We will sharpen this point in the revision by positioning SIGS as a method for structured analytical discovery under admissible priors, and by stating more explicitly that automated Ansatz selection is a natural next step.

---

### Decision · Program_Chairs · 2026-04-30

**Decision:**

Accept (regular)

**Comment:**

The submission has clear strengths with its technically coherent neuro-symbolic method, strong performance on the chosen benchmark suite, and interesting results on coupled nonlinear systems. At the same time, reviewers were skeptical about its dependence on substantial human "input" such as the Ansatz, grammar, and atom library. I do believe this does call for a slight repositioning and a more carefully scoping and framing structured symbolic discovery under selected priors rather than open-ended automated PDE solving. I also agree that the empirical support for generality would be stronger with cleaner ablations disentangling the role of latent-manifold search from the manually imposed restrictions more detailed comparisons with broader symbolic regression work. All things considered, I still consider this a well executed and sufficiently novel and interesting contribution assuming the final version incorporates the improvements described in the rebuttal.